# How to Deep-Learn the Theory behind Quark-Gluon Tagging

Sophia Vent[1,2], Ramon Winterhalder[3], and Tilman Plehn[1,4]

**1** Institut für Theoretische Physik, Universität Heidelberg, Germany
**2** Dipartimento di Fisica e Astronomia, Universitá di Bologna, Italy
**3** TIFLab, Universitá degli Studi di Milano & INFN Sezione di Milano, Italy
**4** Interdisciplinary Center for Scientific Computing (IWR), Universität Heidelberg, Germany

July 30, 2025

## Abstract

**Jet taggers provide an ideal testbed for applying explainability techniques to powerful ML tools. For theoretically and experimentally challenging quark-gluon tagging, we first identify the leading latent features that correlate strongly with physics observables, both in a linear and a non-linear approach. Next, we show how Shapley values can assess feature importance, although the standard implementation assumes independent inputs and can lead to distorted attributions in the presence of correlations. Finally, we use symbolic regression to derive compact formulas to approximate the tagger output.**

# 1 Introduction

Whenever we apply modern machine learning (ML) for fundamental physics [1], the same question arises: *Can we extract relevant and interesting physics from trained neural networks?*[*] In principle, this should be possible, given that neural networks outperform algorithmic approaches using low-level detector outputs. Interpreting such transformative ML tools would then give us scientific insights and build trust in their theoretical and experimental robustness. We demonstrate how this can be achieved using concepts from explainable AI (XAI), which involves probing the internal representations of trained, high-performance networks.

We take this XAI step by analyzing the inner workings of ParticleNet [2], a classic architecture for jet tagging. It represents modern ML tools operating on low-level detector inputs, such as jet constituent four-vectors or calorimeter images [3]. They have led to significant advances for jet tagging, including advanced transformer implementations [4–9]. Specific tasks include quark-gluon tagging [10–22], top tagging [23–27], W/Z tagging [28–33], and bottom/charm identification [34–37]. Complementing the classification performance, we look into the trained network and ask:

1. What features does the network rely on?
2. Are these features reliable and robust?
3. Do they align with known key observables?
4. Can formulas approximate the network?

As a testbed, we choose quark-gluon (QG) tagging [38–41]. While practically extremely promising for many LHC analyses, such as separating signal from background in weak boson fusion or mono-jet searches, QG tagging is theoretically and experimentally tricky. The question of whether a jet originates from a quark or gluon is ill-defined beyond leading order and sensitive to soft and collinear splittings. Furthermore, it strongly depends on the parton shower, hadronization, and detector effects [42, 43]. Very generally, gluon jets radiate more than quark jets because of the color charges, $C_F = 4/3 < C_A = 3$, so their increased particle multiplicity scales with the ratio of color factors, known as Casimir scaling [44, 45]. For additional discriminative power, we want to add more observables with different behavior.

The theoretical and experimental subtleties make QG tagging a particularly compelling case for XAI. Because it lacks a clear-cut ground truth and involves nuanced physics, interpretability is not just a bonus but a necessity. We ultimately envision applying such techniques to networks trained on data, where explainability can drive scientific discovery. Meanwhile, simulation-based studies like ours provide a controlled environment for developing and evaluating XAI tools. While not yet fully established, there is a growing number of physics applications employing promising XAI methods, including Shapley values [46–52], symbolic regression [53–61], and other techniques [62–68].

In this work, we analyze the learned ParticleNet representations after training on QG tagging. We aim to determine whether the network rediscovers known physics, uncovers novel discriminative features, or encodes non-trivial correlations that challenge our intuition. First, we introduce the dataset and describe the ParticleNet architecture used for QG tagging in Sec. 2. In Sec. 3, we analyze the latent feature space of the network using linear and nonlinear dimensionality reduction techniques and investigate how the learned features correlate with known jet observables. In Sec. 4, we perform a Shapley value analysis and discuss its benefits and limitations. Finally, in Sec. 5, we apply symbolic regression to construct compact formulas in terms of the leading features.

---

[*]We refrain from using this question as the paper title because of Hinchliffe's rule.

## 2 Dataset and classifier network

Distinguishing quark-initiated from gluon-initiated jets is a long-standing challenge in LHC physics. It can enhance precision in Standard Model (SM) measurements and improve sensitivity in searches for Beyond Standard Model (BSM) physics, where signal and background processes often differ in jet flavor composition. While quark and gluon jets arise from massless QCD splittings, their internal structures differ because of the gluon's larger color charge. It results in higher particle multiplicities and broader radiation patterns. Beyond these qualitative properties, we investigate the performance of ML-based quark-gluon taggers using precision simulations.

Our primary dataset is generated using Pythia 8.2 [69–71] with default tunes and parton shower settings for the parton-level processes

$$q\bar{q} \rightarrow Z(\rightarrow \nu\bar{\nu}) + g \qquad \text{and} \qquad qg \rightarrow Z(\rightarrow \nu\bar{\nu}) + (\text{uds}) \,. \tag{1}$$

As the neutrinos remain undetected, these processes provide a clean quark-gluon jet sample, allowing us to investigate any subtle differences in the jet substructure. In Fig. 1, we show some examples of LO Feynman diagrams for these processes. Later in our analysis in Sec. 3, we compare results using a similar dataset generated with Herwig 7.1 [72,73] to assess robustness across different generators.

Each dataset consists of 2M jets, with up to 100 constituents per jet. We focus on light-flavor jets, and exclude events containing charm or bottom quarks. The jet reconstruction uses the anti-$k_T$ algorithm [74] with $R = 0.4$, implemented in FastJet [75]. We select a subset of 600k jets with a training/validation/test split of 400k/100k/100k, each with a 50:50 mixture of quark and gluon jets.

**Low-level classifier**

Historically, quark-gluon taggers have relied on high-level observables motivated by QCD. Modern, low-level ML taggers, such as ParticleNet [2], operate on raw information about jet constituents, allowing the network to learn discriminative patterns without any bottleneck. This paradigm shift raises an important question: Are the features learned by such networks consistent with established high-level observables, or do they capture more complex, possibly novel, correlations in the data?

To address this, we examine the internal representations learned by ParticleNet. We aim to determine whether the network implicitly reconstructs established observables, identifies new combinations of known features, or encodes latent patterns that are difficult to interpret. In subsequent sections, we analyze the structure of the latent space, explore its correlation with physics-motivated observables, and investigate the minimal set of features necessary to preserve full classification performance.

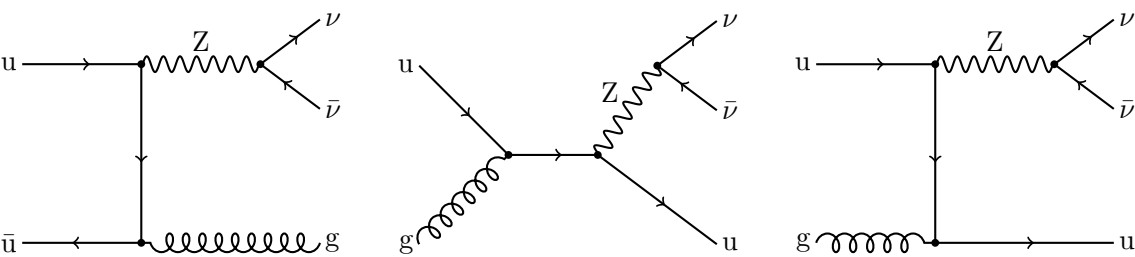

Figure 1: Examples of LO Feynman diagrams leading to gluon and quark jets.

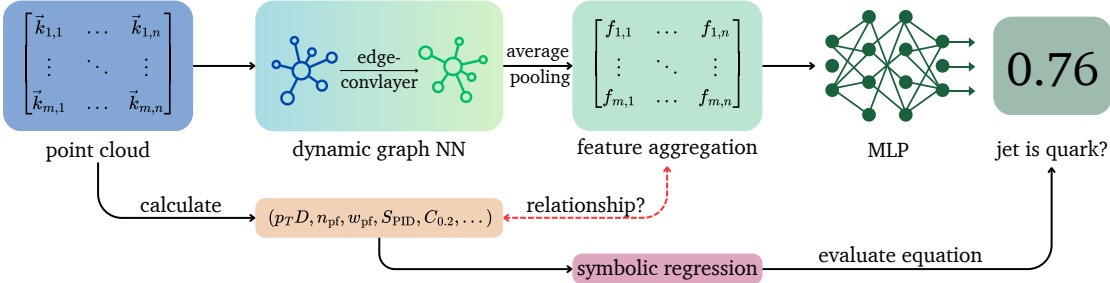

Figure 2: Overview of the ParticleNet architecture and its connection to the explainability techniques explored in our study.

## ParticleNet

ParticleNet [2] is a graph convolutional network that processes unordered sets of jet constituents. Each constituent $i$ is represented by a set of low-level features, such as angular distances from the jet axis, momentum, energy, and particle identification (PID). The full set of input features we will use is

$$\left\{ \Delta\eta_i, \Delta\phi_i, \Delta R_i, \ \log p_{T,i}, \log \frac{p_{T,i}}{p_{T,\mathrm{jet}}}, \ \log E_i, \log \frac{E_i}{E_{\mathrm{jet}}}, \ \mathrm{PID}_i \right\} . \tag{2}$$

Following the ParticleNet convention, we only consider five different particle categories: electrons, muons, charged hadrons, neutral hadrons, and photons. The electric charge is included in the feature set, consistent with the original ParticleNet design, and we encode the PID by a one-hot encoding.

The features in Eq. (2) are passed through a series of edge convolution (EdgeConv) layers. At each layer $l$, the network constructs a dynamic graph by connecting each particle $i$ to its $k$-nearest neighbors $j \in \mathcal{N}(i)$ in the learned feature space. The per-particle feature vector $h_i^{(l)}$ is then updated using learned pairwise interactions:

$$h_i^{(l+1)} = \sum_{j \in \mathcal{N}(i)} f^{(l)}\left(h_i^{(l)}, h_j^{(l)} - h_i^{(l)}\right) , \tag{3}$$

where $f^{(l)}$ denotes a sub-network at layer $l$. This formulation allows the network to learn local patterns and update the particle features accordingly. At the end, per-particle features are aggregated using average pooling to produce a fixed-size jet representation, which is then passed through a multilayer perceptron (MLP) to output a binary classification probability.

Figure 2 illustrates the overall structure of the ParticleNet classifier alongside the inputs and outputs used in our explainability analysis. While the upper path corresponds to the standard inference pipeline described above, the lower path highlights how high-level observables derived from the point cloud can serve as inputs to methods such as symbolic regression or Shapley-based feature attribution. These techniques allow us to probe which physically motivated features the tagger may be implicitly relying on.

We use the compact ParticleNet-Lite variant. It utilizes a single, smaller edge convolution block and outputs a 64-dimensional pooled feature vector per jet. It simplifies the full ParticleNet architecture, which employs two edge convolution blocks and produces a 256-dimensional feature vector.

## 3  From latent features to observables

ParticleNet-Lite learns a 64-dimensional representation of each jet, but it is not obvious that they are all needed for classification. Compressing the representation is the first step towards explainability. We extract the output of the average pooling layer from ParticleNet-Lite, a 64-dimensional vector summarizing each jet, and study its structure using a linear principal component analysis (PCA) and a latent representation from an autoencoder.

### 3.1  Linear correlations

PCA [76] reduces the dimensionality of data by identifying directions that maximize variance. Let

$$X \in \mathbb{R}^{N \times d} \qquad \text{with} \qquad d = 64 \tag{4}$$

be the features of $N$ jets. After zero-centering each feature, we compute the empirical covariance matrix

$$\Sigma = \frac{1}{N-1}(X - \mu_X)^\top (X - \mu_X) \,. \tag{5}$$

We then perform an eigen-decomposition

$$\Sigma = V \Sigma_0 V^\top \,, \tag{6}$$

where $\Sigma_0$ is diagonal, the eigenvalues are called explained variances, i.e. the leading principal components (PCs) capture directions of maximal variance in the latent space. The matrix $V$ gives the principal directions as eigenvectors, so in the PC basis the jet data is given by

$$Z = XV \,. \tag{7}$$

To evaluate the impact of the PCs for classification, we train a simple quark-gluon classifier on a set of leading $k$ PCs and determine their AUCs. This tells us how much discriminative power can be retained in lower-dimensional representations. In Fig. 3 we see that the first five principal components are sufficient to recover the ParticleNet-Lite performance, AUC = 0.902. The leading three PCs already yield an AUC > 0.89.

To ensure resilience, we repeat the analysis using a `Herwig` dataset and show similar results also in Fig. 3. Even when the PCA transformation is learned on `Pythia` jets and applied to `Herwig` jets, the performance remains comparable. This suggests that the principal directions are relatively universal across generators. Altogether, the performance degrades when using `Herwig`, relative to `Pythia`, consistent with previous results [43].

To understand the compressed latent space learned by ParticleNet, we compare the leading PCs with standard substructure observables like the particle multiplicity $n_{\text{pf}}$, the first radial moment or girth $w_{\text{pf}}$ [77,78], the two-point energy correlation function $C_\beta$ for $\beta = 0.2$ [79], and the width of the $p_T$-distribution of the constituents $p_T D$,

$$n_{\text{pf}} = \sum_i 1 \qquad\qquad w_{\text{pf}} = \frac{\sum_i p_{T,i} \Delta R_{i,\text{jet}}}{p_{T,\text{jet}}}$$

$$C_\beta = \frac{\sum_{i<j} p_{T,i} p_{T,j} (\Delta R_{ij})^\beta}{\left(\sum_i p_{T,i}\right)^2} \qquad\qquad p_T D = \frac{\sqrt{\sum_i p_{T,i}^2}}{\sum_i p_{T,i}} \,. \tag{8}$$

We investigate how they are correlated with the three leading principal components of our trained quark-gluon tagger.

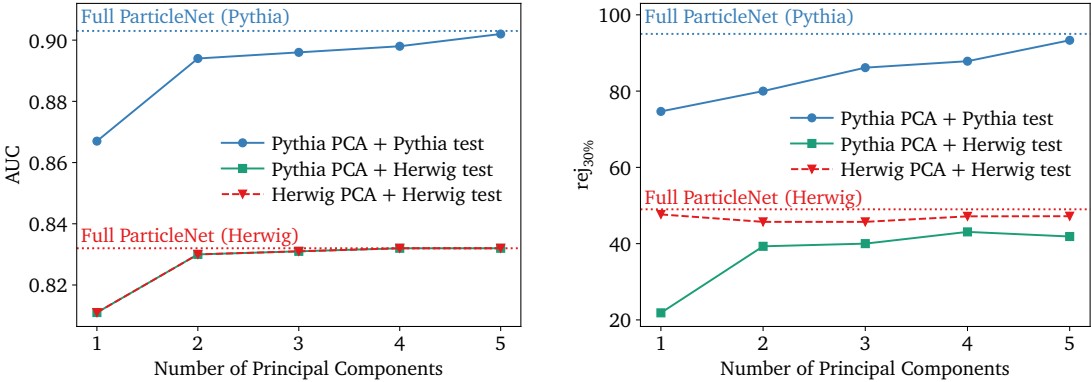

Figure 3: Performance of a NN-classifier on the best-performing PC combinations compared to the full ParticleNet-Lite for `Herwig` and `Pythia` data sets based on AUC and the rejection rate at 30% efficiency.

### PC$_1$: constituent number and diversity

In Fig. 4, we see that the first principal component PC$_1$ is dominated by observables related to the number of particles and their particle nature. Two strongly correlated observables with PC$_1$ are $n_{\mathrm{pf}}$ and the charged multiplicity $n_Q$, defined as the number of charged particles within a jet. In addition, PC$_1$ is correlated with the PID entropy

$$S_{\mathrm{PID}} = - \sum_{\mathrm{type}\, j} f_j \log f_j \,, \tag{9}$$

where $f_j$ is the fraction of particles of type $j$. It captures the diversity of particle types in the jet. Gluon jets, which radiate more and produce a broader mix of particles, have larger $S_{\mathrm{PID}}$ and multiplicity. The linear combination

$$n_{\mathrm{pf}} + \alpha \cdot S_{\mathrm{PID}} = n_{\mathrm{pf}} + 28.6 \cdot S_{\mathrm{PID}}, \tag{10}$$

achieves a slightly higher correlation with PC$_1$. The factor 28.6 was tuned to maximize the correllation with PC$_1$. This indicates that $S_{\mathrm{PID}}$ adds information and is not just correlated with PC$_1$ through $n_{\mathrm{pf}}$. This means PC$_1$ reflects both the quantity and diversity of jet constituents.

### PC$_2$: Radial Energy Profile

Also in Fig. 4, we see that PC$_2$ captures how energy is distributed radially around the jet axis, independently of multiplicity. It is correlated with several observables sensitive to jet width and shape. An observable only correlated with PC$_2$ is the ellipticity, defined in terms of the jet inertia tensor [80] in the transverse plane,

$$I^{ij} = \sum_{k \in \mathrm{jet}} p_{T,k} \frac{r_k^i r_k^j}{r_k^2} \,, \tag{11}$$

Here, $r_k^i$ are the components of the transverse position vector of the constituent $k$ relative to the jet axis. The ellipticity is given in terms of $\xi_{\min}$ and $\xi_{\max}$ as the eigenvalues of this tensor,

$$\epsilon = \frac{\xi_{\min}}{\xi_{\max}} \,. \tag{12}$$

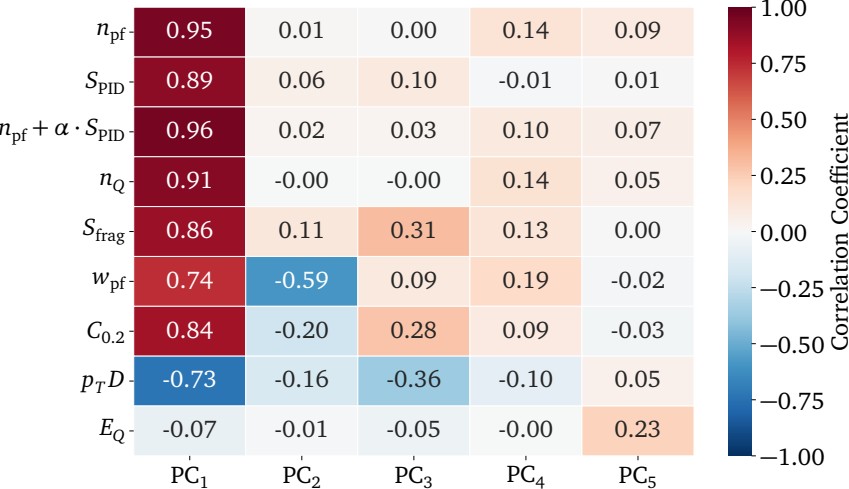

Figure 4: Correlation between the PCs and observables related to multiplicity and particle-type diversity as well as standard observables.

Lower ellipticity corresponds to more elongated (non-circular) jets. Furthermore, $PC_2$ is strongly correlated with $w_{pf}$, which is in turn correlated with $n_{pf}$ and $PC_1$. To exploit this additional direction, we construct the de-correlated combination

$$w_{pf}^{\perp} = \alpha \cdot n_{pf} - w_{pf} \,, \tag{13}$$

where $\alpha = 0.0016$ minimizes the linear correlation with $n_{pf}$. It remains sensitive to the jet width while removing the dependence on the multiplicity. The minus sign is chosen since $w_{pf}$ is negatively correlated with $PC_2$ and we chose to obtain a positive correlation. In addition, we introduce the generalized angularities [42]

$$\lambda_k^{\beta} = \sum_i z_i^{\beta} \Delta R^k \qquad \text{with} \qquad z_i = \frac{p_{T,i}}{p_{T,\text{jet}}} \,. \tag{14}$$

Of them, $\lambda_1^2$ is strongly correlated with $PC_2$. In the spirit of energy correlation functions, we can define a ratio between the Les Houches angularity $\lambda_{0.5}^1$ [42, 81] and $\lambda_1^2$

$$r_\lambda = \frac{\lambda_{0.5}^1}{\lambda_1^2} \,. \tag{15}$$

The numerator $\lambda_{0.5}^1$ gives weight to soft emissions at moderate angular scales, typical for gluon jets; the denominator normalizes the broader radial energy profile. This construction has several advantages: (i) it captures the core and the periphery of the jet; (ii) it is dimensionless and robust against global energy rescaling; and (iii) it is naturally decorrelated from the multiplicity as the numerator and denominator share the same structure. The left panel of Fig. 5 shows that $PC_2$ captures the genuine jet shape and radial flow, distinct from $PC_1$.

**$PC_3$: fragmentation and energy dispersion**

Finally, $PC_3$ is associated with the way energy is shared among jet constituents, corresponding to the fragmentation pattern. The fragmentation entropy is given by

$$S_{\text{frag}} = -\sum_i z_i \log z_i \,, \tag{16}$$

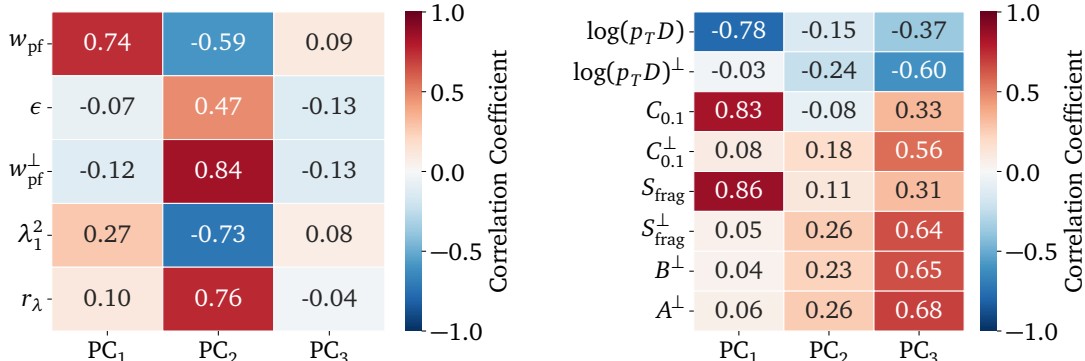

Figure 5: Left: correlation between $PC_2$ and radial jet shape observables. Right: correlations between $PC_3$ and fragmentation and energy dispersion observables.

and measures how evenly the transverse momentum is distributed. Quark jets tend to have a lower fragmentation entropy because of their harder fragmentation.

The PCA components are linearly uncorrelated, so $PC_3$ captures variation in the latent space orthogonal to $PC_1 \approx n_{pf}$ and $PC_2 \approx w_{pf}^\perp$. This suggests that $PC_3$ encodes physical information that is independent of these two observables and is instead related to fragmentation and energy dispersion. To test this, we construct candidate observables that may align with $PC_3$, but remove any linear correlation with $n_{pf}$ and $w_{pf}^\perp$,

$$O^\perp = O - \beta n_{pf} - \gamma w_{pf}^\perp. \tag{17}$$

where $\beta$ and $\gamma$ are chosen to minimize the correlation with $n_{pf}$ and $w_{pf}^\perp$. This isolates the part of $O$ that corresponds to the latent direction of $PC_3$. After testing various combinations, the following observables are strongly correlated with $PC_3$:

$$A^\perp = S_{frag} \frac{C_{0.1}}{C_{0.05}} - 0.03 \cdot n_{pf} + 1.95 w_{pf}^\perp$$
$$B^\perp = -C_{0.1} \cdot \log(p_T D) \cdot C_{0.05} - 0.014 n_{pf} + 21.32 w_{pf}^\perp$$
$$S^\perp = S - 0.03 n_{pf} + 0.45 w_{pf}^\perp$$
$$C_{0.1}^\perp = C_{0.1} - 0.0046 n_{pf} + 0.7701 w_{pf}^\perp$$
$$\log(p_T D)^\perp = \log(p_T D) + 0.0143 \cdot n_{pf} - 0.065 \cdot w_{pf}^\perp. \tag{18}$$

All these observables are sensitive to the distribution of transverse momentum within the jet, characterizing the extent to which the fragmentation pattern is hard or diffuse. In the right panel of Fig. 5 we see that $PC_3$ is significantly correlated with these fragmentation-sensitive observables, which means $PC_3$ capturing aspects of fragmentation dynamics and energy dispersion.

## $PC_4$ and $PC_5$

Beyond the first three PCs, a clear physical interpretation becomes increasingly challenging. This difficulty arises because many jet observables are highly correlated and tend to span similar directions in feature space. Consequently, the leading PCs capture most of the variance associated with well-understood QCD observables, while the subleading components reflect more subtle structures that may represent combinations of multiple physical effects.

In our analysis, we did not find an individual observable that is strongly or uniquely correlated with $PC_4$. However, $PC_5$ shows a notable correlation with the charged energy fraction

$$E_Q = \frac{E_{\text{charged}}}{E_{\text{jet}}} \, . \tag{19}$$

This suggests that the ParticleNet learns charge-related information in a non-trivial and decorrelated way. Unlike $PC_{1-3}$, which align closely with standard observables, $PC_5$ does not map directly onto a single feature but captures a more subtle charge structure of the jet.

## 3.2 Non-linear correlations

Using PCA we have analyzed the latent space of ParticleNet-Lite in a linearized approximation. To probe non-linear structures we introduce a Disentangled Latent Classifier (DLC), a network that compresses the 64-dimensional $X$ into a lower-dimensional latent representation, while simultaneously learning to classify quark and gluon jets. The architecture is visualized in Fig. 6. The DLC consists of three components: (i) an encoder that maps each input $x_i \in \mathbb{R}^{64}$, drawn from the dataset $X \in \mathbb{R}^{N \times 64}$, to a latent vector $z_i \in \mathbb{R}^d$; (ii) a decoder that reconstructs the original input $\hat{x}_i$ from $z_i$; and (iii) a classification head that predicts the jet label $y_i \in \{0, 1\}$ from the latent representation. The corresponding loss for $N$ jets is

$$\mathcal{L} = \underbrace{\frac{1}{N} \sum_{i=1}^{N} |x_i - \hat{x}_i|^2}_{\mathcal{L}_{\text{reco}}} + \underbrace{\frac{1}{N} \sum_{i=1}^{N} \Big[ y_i \log \sigma(z_i) + (1 - y_i) \log(1 - \sigma(z_i)) \Big]}_{\mathcal{L}_{\text{class}}}$$
$$+ \underbrace{\sum_{j \neq k} \Big[ \text{Cov}(z_j, z_k) \Big]^2}_{\mathcal{L}_{\text{disentangle}}} \, . \tag{20}$$

The first term is the MSE between the input vector $x_i$ and its reconstruction $\hat{x}_i$. The second term is the binary cross-entropy, where $\sigma(z_i)$ denotes the predicted probability for jet $i$ in its latent representation. Crucially, the third term penalizes correlations between components of the latent vector by summing the squared off-diagonal elements of the covariance matrix. This forces the network to encode independent features in each latent direction.

To determine the minimal latent dimensionality for successful classification, we train the DLC with different latent space sizes and evaluate the AUC and the calibration. Calibration

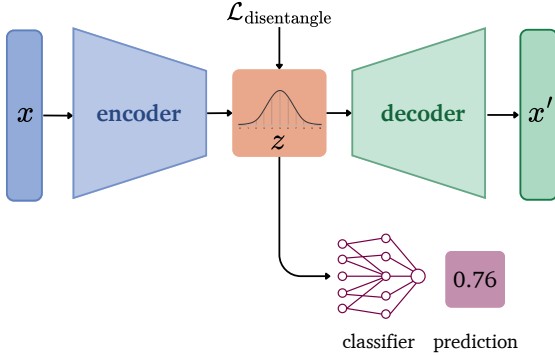

Figure 6: Setup of the DLC. The input is compressed through the encoder to a latent space $z$, which is enforced to be disentangled. The disentangled latent space is then passed to a classifier as well as a decoder for reconstruction

| Latent Dim | 1 | 2 | 3 | 4 |
|---|---|---|---|---|
| AUC | 0.893(2) | 0.9001(4) | 0.9024(4) | 0.9034(2) |
| $\text{rej}_{30\%}$ | 72(3) | 77(3) | ´95(5) | 95(3) |
| $\Delta C$ | 1.8(3) | 0.93(5) | 1.0(16) | 0.9(15) |

Table 1: AUC and calibration for different latent dimensions, averaged over five runs.

curves test how well predicted probabilities match observed frequencies. A perfectly calibrated network produces a diagonal curve. To quantify the deviation from the diagonal we use the expected calibration error over $K$ bins of the calibration curve

$$\Delta C = 100 \cdot \sum_{k=1}^{K} \frac{N_k}{K} |p_k - f_k| \,, \tag{21}$$

where $N_k$ are the number of events in the $k^{\text{th}}$ bin, $p_k$ is the average predicted probability in that bin, and $f_k$ is the observed frequency of positive labels. A lower $\Delta C$ means better calibration. For readability we include a factor of 100.

Table 1 shows that a latent space with just three dimensions achieves nearly the same AUC as the full 64-dimensional ParticleNet-Lite output, as well as a decent calibration. In principle, any classifier maps to a scalar output and constructs a one-dimensional representation for discrimination. However, including a reconstruction loss constrains the network to preserve a compressed version of the full feature space, yielding a structured and interpretable latent representation.

To assess the physical meaning of the latent dimensions, we examine correlations between latent variables and jet observables in Fig. 7. We find a structure very similar to the PCA analysis. The first latent dimension is dominated by $n_{\text{pf}}$ and $S_{\text{PID}}$, the second by shape observables, and the third by fragmentation observables.

While the latent dimensions in DLC are learned to be uncorrelated, their separation is less clean than in the PCA. This is because the nonlinear transformations can cause overlap in the physical interpretation of different latent directions. For instance, $\log(p_T D)$ is strongly correlated with the first $z_1$, but this correlation disappears once the linear dependence on $n_{\text{pf}}$ is removed. This suggests that the correlation is largely due to the strong dependence of

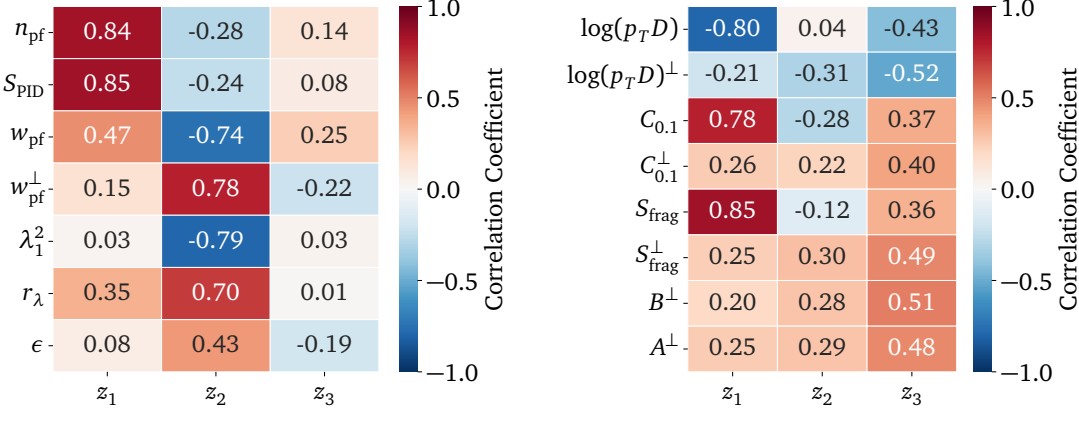

Figure 7: Correlations between physics observables and the disentangled latent spaces $z_i$.

$\log(p_T D)$ on multiplicity, rather than an intrinsic feature of $z_1$. While the DLC structure resembles PCA, the mapping between latent dimensions and physical observables is more general but less direct.

# 4 Feature importance from Shapley values

Rather than learning and analyzing latent representations, we can train a simple NN-classifier and analyze the feature or observable importance using the SHapley Additive exPlanations (SHAP) framework [82]. Shapley values assign a contribution to each input feature for the classifier output $f(x)$, based on cooperative game theory [83]. For a given feature $i$, the Shapley value $\mathcal{V}_i$ is defined as the average marginal contribution of $i$ across all subsets of the remaining features:

$$\mathcal{V}_i = \sum_{S \subseteq F \setminus \{i\}} \frac{|S|!(|F| - |S| - 1)!}{|F|!} \Big[ f(S \cup \{i\}) - f(S) \Big]. \tag{22}$$

Here, $F$ is the full set of input features (e.g. jet observables), and $S$ is a subset of $F$ that does not contain $i$. The term $|S|$ ($|F|$) denotes the number of features in $S$ ($F$), and the sum averages the contribution of feature $i$ over all such subsets. The model output $f(S)$ represents the expected classifier prediction when only the features in $S$ are known. Computing $f(S)$ requires marginalizing over the remaining features $B = F \setminus S$, which is generally intractable. To make this feasible, the model-agnostic kernel SHAP makes a simplifying assumption and replaces the conditional distribution $p(x_B | x_S)$ with the marginal distribution $p(x_B)$:

$$f(S) = \big\langle f(x_S, x_B) \big\rangle_{x_B \sim p(x_B | x_S)} \approx \big\langle f(x_S, x_B) \big\rangle_{x_B \sim p(x_B)}. \tag{23}$$

It renders the Shapley analysis numerically feasible, but it can lead to misleading attributions when features are strongly correlated. In such cases, SHAP may undervalue features that are informative but share mutual information with others.

Table 2 shows AUC scores for various combinations of high-level observables, to guide our choice of input sets for SHAP analysis. The left panel of Fig. 8 shows the SHAP summary for the

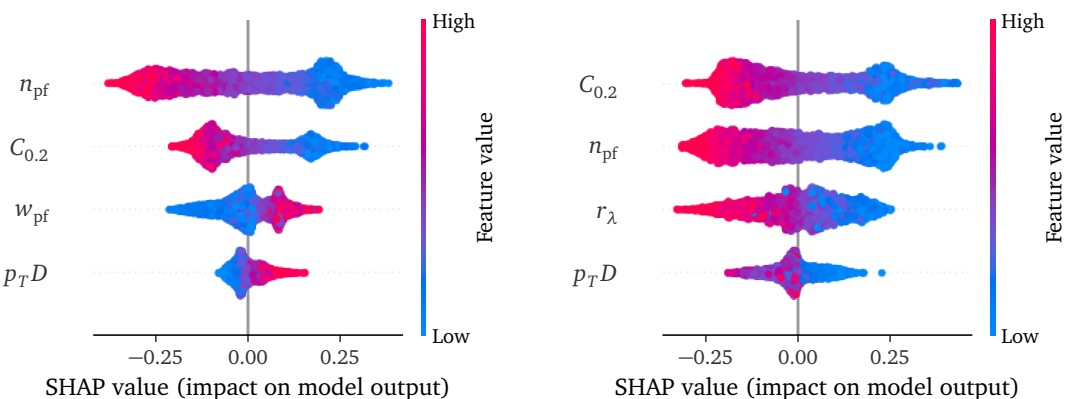

Figure 8: SHAP summaries for the set of observables given in Eq.(8) (left), and after replacing $w_{\text{pf}}$ with $r_\lambda$ (right). Features are ranked by average importance from red (high) to blue (low).

| interpretation | Observables | AUC | $\text{rej}_{30\%}$ | $\text{rej}_{50\%}$ |
|---|---|---|---|---|
| standard set | $n_{\text{pf}}, w_{\text{pf}}, p_T D, C_{0.2}$ | 0.8626 | 71.11 | 21.53 |
| decorrelated girth | $n_{\text{pf}}, r_\lambda, p_T D, C_{0.2}$ | 0.8720 | 70.00 | 23.33 |
| PC1 approx. | $n_{\text{pf}}$ | 0.8406 | 53.33 | 16.65 |
| | $S_{\text{PID}}$ | 0.8402 | 61.37 | 19.31 |
| | $n_{\text{pf}}, S_{\text{PID}}$ | 0.8471 | 61.37 | 20.00 |
| PC1-2 approx. | $n_{\text{pf}}, S_{\text{PID}}, w_{\text{pf}}$ | 0.8629 | 70.00 | 20.64 |
| | $n_{\text{pf}}, S_{\text{PID}}, w_{\text{pf}}^\perp$ | 0.8632 | 68.92 | 20.64 |
| | $n_{\text{pf}}, S_{\text{PID}}, r_\lambda$ | 0.8703 | 64.00 | 20.74 |
| PC1-3 approx. | $n_{\text{pf}}, r_\lambda, A^\perp$ | 0.8645 | 67.88 | 19.31 |
| | $n_{\text{pf}}, S_{\text{PID}}, r_\lambda, A^\perp$ | 0.8725 | 70.00 | 21.13 |
| | $n_{\text{pf}}, S_{\text{PID}}, r_\lambda, C_{0.2}$ | 0.8728 | 73.44 | 22.40 |
| | $n_{\text{pf}}, S_{\text{PID}}, r_\lambda, C_{0.2}, p_T D$ | 0.8777 | 77.24 | 24.34 |
| | $n_{\text{pf}}, S_{\text{PID}}, r_\lambda, C_{0.2}, p_T D, S_{\text{frag}}$ | 0.8793 | 74.66 | 24.63 |
| PC1-5 approx. | $n_{\text{pf}}, S_{\text{PID}}, r_\lambda, S_{\text{frag}}, C_{0.2}, p_T D, E_Q$ | 0.8806 | 80.00 | 25.60 |

Table 2: AUC scores for different combinations of three to seven high-level jet observables.

standard tagging observables defined in Eq.(8). Positive SHAP values indicate that a feature increases the confidence of the network in the quark label, negative values push the prediction toward the gluon label. The features $n_{\text{pf}}$ and $C_{0.2}$ behave as expected: low particle multiplicity and a small energy correlation suggest a quark jet. We also see that $p_T D$ contributes little to the classification.

The feature $w_{\text{pf}}$ in the same panel displays a counter-intuitive pattern: jets with low $w_{\text{pf}}$, typically indicative of narrow, quark-like jets, receive negative SHAP values related to a gluon classification. This is not a failure of the classifier, but a limitation of the SHAP attribution mechanism. Low $w_{\text{pf}}$ occurs in both, quark jets (with low $n_{\text{pf}}$) and some gluon jets (with high $n_{\text{pf}}$), due to their correlation. The classifier correctly learns that low $w_{\text{pf}}$ combined with high $n_{\text{pf}}$ is characteristic of gluon jets. However, SHAP evaluates the contribution of $w_{\text{pf}}$ by marginalizing over $n_{\text{pf}}$ and other features, assuming independence and thereby ignoring their joint structure. As a result, SHAP assigns a negative contribution to $w_{\text{pf}}$ even when, conditional on $n_{\text{pf}}$, it should favor a quark classification.

To address this mis-attribution, we replace $w_{\text{pf}}$ with $r_\lambda$, the decorrelated alternative introduced in Eq.(15), as part of a minimal input set,

$$\left\{ n_{\text{pf}}, w_{\text{pf}}, p_T D, C_{0.2} \right\} \quad \longrightarrow \quad \left\{ n_{\text{pf}}, r_\lambda, p_T D, C_{0.2} \right\} . \tag{24}$$

In the right panel of Fig. 8 we can now see a clear interpretation of the features for $r_\lambda$, $n_{\text{pf}}$ and $C_{0.2}$. SHAP estimates the effect of each input by averaging over all possible subsets, assuming that the features are independent. This is why now $p_T D$ shows a similar correlation problem as $w_{\text{pf}}$ did before.

This issue becomes even clearer when we include more features. Figure 9 shows SHAP values for a network trained on inputs that approximate the first three principal components from Sec. 3. Here, the apparent importance of $n_{\text{pf}}$ is suppressed, not because it's actually unimportant, but because it's hard to separate its effect when the features are strongly correlated. Changing $n_{\text{pf}}$ inevitably changes the other observables, which means that the overall impact on the classifier is larger than what SHAP picks up when assuming independence.

If we just look at these SHAP plots, we might wrongly conclude that $n_{\text{pf}}$ has only moderate

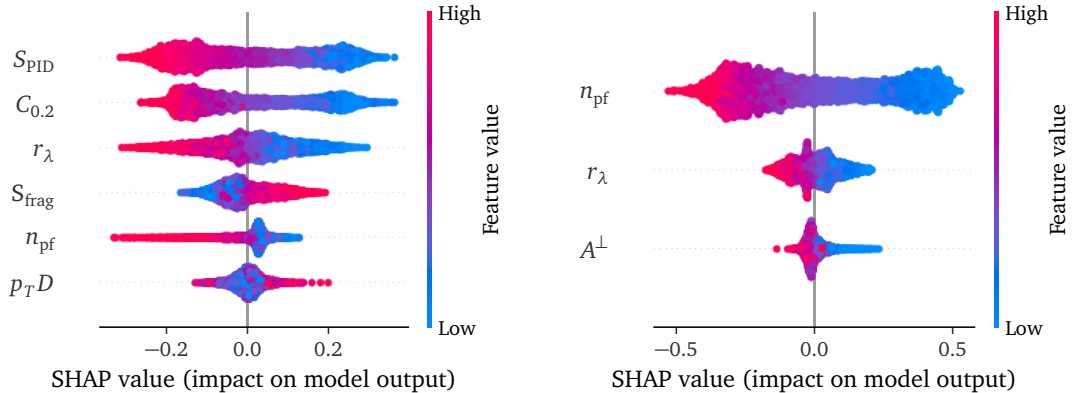

Figure 9: SHAP summaries for set of observables approximating the first three PCA components. Left: strong correlations among features distort SHAP attributions, particularly for $n_{\mathrm{pf}}$ where the importance is underestimated. Right: SHAP values for a decorrelated feature set approximating the first three PCA components. The importance of each feature aligns with the ranking of principal components.

influence. This goes against both our PCA findings and what we expect from QCD. This means that SHAP needs to be interpreted carefully: marginalizing over correlated features does not capture the real joint behavior. To avoid this, we build a decorrelated feature set that roughly tracks the first three PCA directions:

$$\{n_{\mathrm{pf}}, r_\lambda, A^\perp\}. \tag{25}$$

We could have swapped $n_{\mathrm{pf}}$ for $S_{\mathrm{PID}}$, or picked other observables for the third component, but we use $A^\perp$ since it combines energy correlations and fragmentation entropy. The SHAP plot for this set is shown to the right in Fig. 9, and it aligns with our earlier PCA findings. $n_{\mathrm{pf}}$ is clearly the leading feature, with $r_\lambda$ and $A^\perp$ adding extra discrimination.

By default, SHAP ranks features by their average absolute contribution. When the inputs are decorrelated, this ranking generally matches both intuition and our PCA results, making it easier to interpret. But with correlated features, SHAP can give misleading priorities, even if the overall classifier still works fine. In practice, this means we need to prepare inputs carefully, using decorrelated features or something like a PCA basis, to get SHAP explanations that actually reflect the physics. SHAP is still a powerful tool, we just have to be mindful of these subtleties when applying it to jet observables.

## 5   Symbolic regression

Having analyzed the internal structure of the trained network using principal components and Shapley values, we now ask directly: Can the classifier output be approximated by a formula built from high-level observables? This question leads directly to the language of theoretical physics, i.e. formulas and equations. In principle, neural networks can be approximated by formulas, and the extremely strong regularization of formulas can be helpful in cases of (too) little training data [61]. Instead of reasoning about latent vectors or weight matrices, we aim to represent the trained ParticleNet as a formula, capturing its dependencies on subjet observables. Our goal is to connect the machine-learned decision boundaries to known QCD

patterns, such as inverse scaling with particle multiplicity or nonlinear dependence on energy sharing and fragmentation.

We perform symbolic regression using the PySR framework [55], which searches for formulas by evolving a population of candidate formulas through a genetic algorithm. Each candidate is represented as a tree, constructed from a predefined set of mathematical operations. Each node in the tree contributes to the complexity. For example, the equation

$$3x + a \tag{26}$$

has a complexity of five, three for the parameters and two for the operations. The algorithm balances two competing objectives, accuracy and simplicity. This makes PySR particularly well suited for discovering compact formulas that approximate the network output while remaining physically interpretable.

**Setup and method**

We first select a set of observables based on their performance and interpretability, as discussed in the previous sections. These include particle multiplicity, radial energy distribution, fragmentation entropy, momentum balance, the two-point correlation function, the charged energy fraction, and the PID entropy,

$$\{n_{\mathrm{pf}}, r_\lambda, S_{\mathrm{frag}}, p_T D, C_{0.2}, E_Q, S_{\mathrm{PID}}\} \tag{27}$$

For each input observable (or combination), we first train a simple neural network classifier that uses only those observables. Its output defines the target for the symbolic regression. This isolates the contribution of the chosen observables and ensures that the formulas approximate a realistic, learnable decision function.

For the symbolic regression, we use PySR with a fixed set of operators including addition, multiplication, division, powers, and a small number of nonlinear activation functions such as $\tanh(x)$. For single-observable fits, we allow a maximum complexity of 10; for two-observable combinations, we increase the limit to 22. The formulas are evaluated based on three criteria that balance precision and interpretability:

1. the area under the ROC curve (AUC);
2. the background rejection at 30% quark efficiency; and
3. the calibration metric $\Delta C$ defined in Eq.(21).

## 5.1 One-dimensional regression

We begin by applying symbolic regression to individual high-level observables, to see whether the tagger's decision surface, restricted to a single input observable, can be captured by a simple formula. For each observable $\mathcal{O}$, we train a neural network using only $\mathcal{O}$ as input and record its predicted quark probability $y_{\mathcal{O}}$. Symbolic regression is then used to approximate

$$f(\mathcal{O}) \approx y_{\mathcal{O}} \,. \tag{28}$$

To understand the role of functional choices, we first focus on the particle multiplicity $n_{\mathrm{pf}}$ as the most discriminative observables for quark-gluon tagging.

**Activation functions**

We first study how the choice of nonlinear activation functions affects the learned formulas. Since the tagger outputs are bounded between 0 and 1, it is natural to include bounded nonlinearities that can model this range effectively. At the same time, we want to keep the formulas

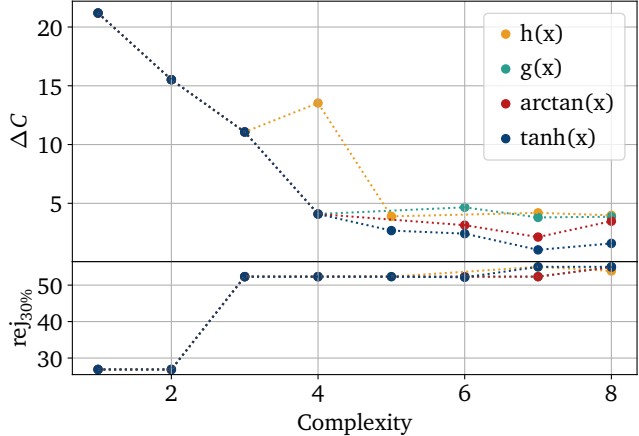

Figure 10: Comparison of activation functions in symbolic regression using $n_{\mathrm{pf}}$. The calibration error $\Delta C$ and background rejection at 30% quark efficiency are shown as a function of formula complexity.

as compact and interpretable as possible. We test standard options like $\tanh(x)$ and $\arctan(x)$, as well as two custom choices

$$h(x) = \frac{-x}{\sqrt{1+x^2}} \qquad \text{and} \qquad g(x) = \frac{1}{\pi}\arctan(x) + 0.5 \,. \tag{29}$$

In principle, PySR can build complex activation functions such as the sigmoid from elementary functions, but this would require a complexity of 19. Hence, explicitly providing more complex activation functions significantly boosts the performance. However, there are two reasons for limiting the symbolic algorithm to a single nonlinear function. First, PySR develops a bias toward an activation function appearing early in its formula search. Once a function like $\tanh(x)$ appears, the evolutionary search typically continues to build on it and ignores the better performance of alternative functions. Second, fixing the activation function reduces the complexity and prevents overly convoluted structures.

To explore this in a controlled setting, we perform a scan using only $n_{\mathrm{pf}}$ as input, and vary the allowed complexity from 1 to 10. For each function, we track the calibration error $\Delta C$ and the background rejection at 30% signal efficiency. Figure 10 summarizes the results.

The different functions perform similarly in terms of AUC, but $\tanh(x)$ consistently produces better-calibrated outputs and leads to shorter, cleaner formulas. Based on this, we adopt $\tanh(x)$ as the default nonlinearity for all remaining regressions. Note that we also considered the sigmoid function in earlier tests, but it was excluded from the final analysis due to its inferior performance.

**Monotonicity and constant AUC**

Looking at Table 3, we see how the formula evolves with complexity. At low complexity, the network starts with a simple inverse scaling, $\sim 1/n_{\mathrm{pf}}$, capturing the trend that higher multiplicities are associated with gluon jets. As complexity increases, PySR sharpens this behavior by adding nonlinear functions like tanh and raising them to higher powers. These refinements do not change the overall monotonic trend, but they improve the match to the classifier output. From complexity 7, additional gains come mainly from fine-tuning the shape. The formulas remain compact and interpretable, with increasing agreement with the network output.

| complexity | equation | loss | AUC | Rej$_{30\%}$ | $\Delta C$ |
|---|---|---|---|---|---|
| 1 | $0.5$ | 0.083 | 0.5 | 3.33 | - |
| 3 | $\frac{17.7}{n_{\text{pf}}}$ | 0.029 | 0.839 | 52.32 | 11.34 |
| 4 | $\tanh \frac{22.3}{n_{\text{pf}}}$ | 0.0169 | 0.839 | 52.32 | 12.23 |
| 5 | $\tanh \frac{850.9}{n_{\text{pf}}^2}$ | 0.0009 | 0.839 | 52.32 | 3.43 |
| 6 | $\tanh^6 \frac{57.447636}{n_{\text{pf}}}$ | 0.0006 | 0.839 | 52.32 | 2.04 |
| 7 | $\tanh \frac{44.19}{\left(0.084 \cdot n_{\text{pf}}+1\right)^3}$ | 0.00026 | 0.839 | 52.32 | 1.64 |
| 8 | $\tanh^3 \left( 681.83 \cdot \left( 0.014 + \frac{1}{n_{\text{pf}}} \right)^2 \right)$ | 0.00025 | 0.839 | 52.32 | 1.58 |
| 9 | $0.94 \cdot \tanh \left( 21036 \cdot \left( 0.005 + \frac{1}{n_{\text{pf}}} \right)^3 \right)$ | 0.00004 | 0.839 | 52.32 | 1.08 |

Table 3: 1D Symbolic regression tables for $n_{\text{pf}}$ only.

A subtle point arises when comparing symbolic regressions based on the AUC. Because the order of the ROC curve is invariant under monotonic transformations [84], formulas that differ substantially in sharpness or calibration will give identical AUC scores. This is evident in most of the one-dimensional regressions, where all formulas are monotonic transformations with identical AUCs. In Fig. 11, we see that visually the formulas differ for $n_{\text{pf}}$ even if the AUC remains the same. Additionally, we observe that higher complexities match the calibrated tagger prediction more closely, and we indeed require higher complexities for a good calibration.

**Formulas for each observable**

Having validated our strategy on $n_{\text{pf}}$, we now extend it to the full set of high-level observables

$$\left\{ n_{\text{pf}}, r_\lambda, p_{\text{T}}D, C_{0.2}, S_{\text{PID}}, S_{\text{frag}}, E_Q \right\} . \tag{30}$$

Following the previous sections, they span different aspects of jet substructure, including multiplicity, angular spread, fragmentation, and charge.

For each observable we first train a classifier and then use PySR to approximate it. Our maximum complexity is 10, to ensure the equations are interpretable. Table 4 summarizes the best formulas for each observable, along with the AUC, background rejection at 30% signal

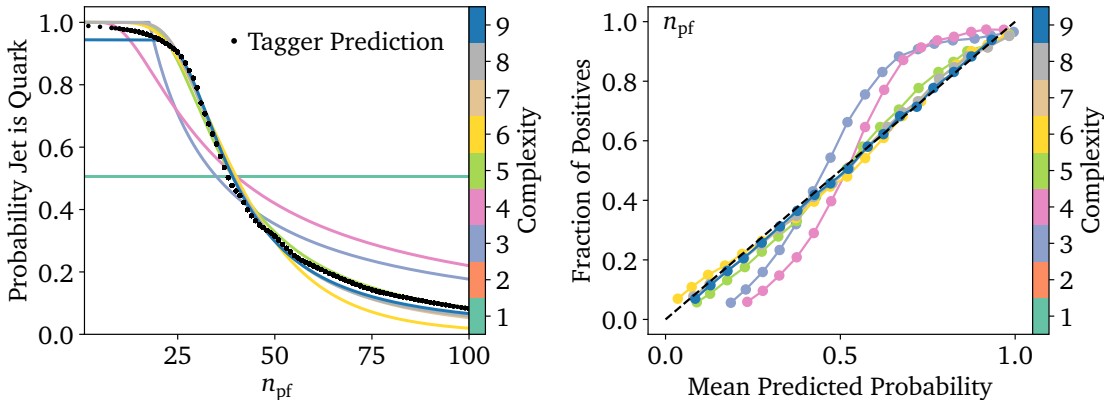

Figure 11: The left panel shows the symbolic regression curves compared to the target tagger prediction. The right panel shows the different calibration curves. A perfect calibration is shown by a diagonal (dashed black line).

| observable | complexity | equation | AUC | Rej$_{30\%}$ | $\Delta C$ |
|---|---|---|---|---|---|
| $S_{\text{PID}}$ | 6 | $\tanh^2 \frac{1.14}{S_{\text{PID}}^3}$ | 0.842 | 61.84 | 1.74 |
| $n_{\text{pf}}$ | 9 | $0.94 \cdot \tanh\left(21036 \cdot \left(0.005 + \frac{1}{n_{\text{pf}}}\right)^3\right)$ | 0.839 | 52.32 | 1.08 |
| $S_{\text{frag}}$ | 6 | $\tanh^2 \frac{18.08}{S_{\text{frag}}^3}$ | 0.828 | 38.97 | 1.80 |
| $p_T D$ | 7 | $0.92 \cdot \tanh\left(19.49 \cdot p_T D^3\right)$ | 0.807 | 26.87 | 1.04 |
| $C_{0.2}$ | 9 | $\tanh\left(343.13 \cdot (0.72 \cdot C_{0.2} - 1)^{18} + 0.22\right)$ | 0.793 | 58.41 | 1.54 |
| $r_\lambda$ | 10 | $\left((0.59 - \tanh(0.0038 \cdot r_\lambda))^2\right)^{0.5} + 0.22$ | 0.637 | 6.46 | 1.81 |

Table 4: Best equations for each observable based on simplicity, performance and calibration. All numbers are rounded to 2 digits for readability.

efficiency, and calibration error $\Delta C$. For each case, we select the simplest formula that achieves good performance across all three metrics. Full complexity scans for each observable are provided in the Appendix.

We can observe patterns in these equations. As expected, $S_{\text{PID}}$ behaves similarly to $n_{\text{pf}}$ and yields a relatively simple formula resembling the inverse scaling predicted by Casimir factors. A higher particle diversity tends to favor the gluon label, which is mapped to 0. On the other end of the spectrum, $r_\lambda$ shows limited discriminative power, and the corresponding formula is noticeably more complex. In general, more informative observables tend to produce simpler formulas, often involving only a few transformations to capture the relevant trends.

## 5.2 Two-dimensional regression

Next, we apply symbolic regression to pairs of observables. We allow for a complexity of 22 to accurately describe the 2-dimensional dependence. Rather than testing all possible combinations, we focus on pairs including $n_{\text{pf}}$ as the standard interpretable observable. To improve numerical stability and keep coefficient scales comparable, we multiply $n_{\text{pf}}$ by a factor of 0.01 before regression. We deliberately avoid automated rescaling or ML-based normalization, as preserving the native physical scales of the observables supports direct interpretability of the resulting formulas.

At low complexities, the formulas remain effectively one-dimensional. This is expected, because even a basic linear combination of two observables can already have a complexity of 7

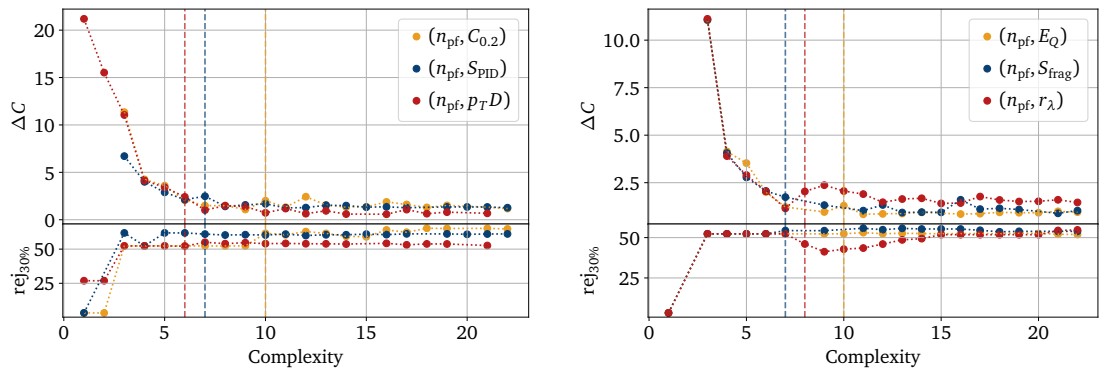

Figure 12: Rejection rates at 30% efficiency and calibration error dependent on the complexity. The vertical lines show when the equation depends for the first time on two observables.

| obs. pair | complexity | equation | AUC | Rej$_{30\%}$ | $\Delta C$ |
|---|---|---|---|---|---|
| $(n_{\mathrm{pf}}, r_\lambda)$ | 15 | $\left(1.1 - n_{\mathrm{pf}}\right) \cdot \tanh\left(181.1 \cdot \left(\sqrt{\frac{1}{r_\lambda}} + \frac{0.003}{n_{\mathrm{pf}}^3}\right)^3\right)$ | 0.860 | 51.81 | 1.41 |
| $(n_{\mathrm{pf}}, C_{0.2})$ | 18 | $\left(0.03 - \tanh^3\left(\frac{82.47 \cdot \left((C_{0.2} - 0.68)^2 - 0.04\right)^2 + 0.54}{n_{\mathrm{pf}}}\right)\right)^2$ | 0.848 | 65.27 | 1.30 |
| $(n_{\mathrm{pf}}, S_{\mathrm{PID}})$ | 11 | $0.94 \cdot \tanh^{0.5}\left(\frac{1.23 \cdot \left(0.90 \cdot n_{\mathrm{pf}} - 1\right)^2}{S_{\mathrm{PID}}^6}\right)$ | 0.847 | 60.96 | 1.25 |
| $(n_{\mathrm{pf}}, S_{\mathrm{frag}})$ | 13 | $0.94 \cdot \tanh^2\left(\frac{0.57}{\left(n_{\mathrm{pf}} - 0.13\right) \cdot \left(n_{\mathrm{pf}} - S_{\mathrm{frag}}\right)}\right)$ | 0.844 | 55.74 | 0.93 |
| $(n_{\mathrm{pf}}, p_T D)$ | 7 | $\tanh \frac{0.26 \cdot p_T D}{n_{\mathrm{pf}}^2}$ | 0.844 | 55.01 | 1.02 |
| $(n_{\mathrm{pf}}, E_Q)$ | 11 | $-0.098 \cdot E_Q + \tanh\left(0.11 + \frac{0.03}{n_{\mathrm{pf}}^3}\right)$ | 0.840 | 52.36 | 0.84 |

Table 5: Symbolic regression results for pairs of observables including $n_{\mathrm{pf}}$. Each formula is selected based on a balance of complexity, AUC, rejection rate, and calibration.

or more. For low complexity thresholds, the second observable is too expensive to be included in the formula. In fact, up to a complexity of 4, the symbolic regression arrives at the same formulas as for $n_{\mathrm{pf}}$ only. The complete set of formulas, ranging from simple to complex, is provided in the Appendix. In Fig. 12 we mark the points at which each second observable begins to appear by vertical lines. Towards higher complexity, the second observables clearly improve the performance. Table 5 summarizes the best formulas for each $n_{\mathrm{pf}}$-based pair, selected by balancing complexity, AUC, rejection rate, and calibration quality.

It is not always obvious which combinations of observables will yield the highest performance gain. Interestingly, $r_\lambda$ performs poorly on its own, but complements $n_{\mathrm{pf}}$ best and leads to an AUC of 0.860. This suggests that it adds information that is decorrelated from $n_{\mathrm{pf}}$. Similarly, $C_{0.2}$ improves the rejection rate, despite its modest individual AUC. In contrast, observables like $p_T D$ and $E_Q$ produce simpler, more readable formulas that still reach competitive performance, especially in terms of calibration. This shows how combining $n_{\mathrm{pf}}$ with a second observable enables symbolic regression to access richer structures, and leads to better discrimination and equally interpretable formulas.

## 5.3 Towards all-observable regression

Finally, we turn to the question if the full ParticleNet can be approximated by a formula in terms of all seven observables from Eq.(27). We already know that adding an additional observable typically increases the formula complexity by at least five. Covering all observables should require a complexity of 40 or more. This scales poorly in terms of computational cost and formula interpretability. To preserve interpretability and remain faithful to the physical meaning of each observable, we avoid standard preprocessing steps such as dataset-based normalization or standardization. However, we do employ global and invertible transformations, for instance, $n_{\mathrm{pf}}$ is rescaled by a factor of 0.01 to prevent large numerical coefficients from dominating the regression. To ensure a fair comparison, we train a new network on the same inputs. In Table 6 we compare the performance of the learned formula to this network.

In contrast to an estimated complexity around 40, we find that a complexity of 22 yields the best trade-off between performance and interpretability. Here, the learned formula matches the performance of the network. The best-performing formula at complexity 22 is

$$\tanh^3\left(0.55 \cdot C_{0.2} + 2 \cdot \left(-0.02 \cdot r_\lambda \cdot \left(C_{0.2} \cdot p_T D \cdot S_{\mathrm{PID}} \cdot S_{\mathrm{frag}} - 0.25\right) + 1\right)^3\right). \tag{31}$$

| observables | model | AUC | Rej$_{30\%}$ |
|---|---|---|---|
| $(n_{\text{pf}}, p_T D, C_{0.2}, r_\lambda, S_{\text{PID}}, S_{\text{frag}}, E_Q)$ | MLP | 0.872 | 66.87 |
| | PySR | 0.871 | 66.58 |

Table 6: Performance of the full model using all observables. The symbolic regression formula is limited to complexity 22.

It involves five of the seven observables, $n_{\text{pf}}$ and $E_Q$ are absent for the given complexity. The formula provides a nontrivial and yet interpretable approximation of the full classifier. The linear term in $C_{0.2}$ provides direct sensitivity to the angular energy distribution, which helps distinguish collimated quark jets from broader gluon jets. The second term encodes a nonlinear interaction between $C_{0.2}$, $p_T D$, $S_{\text{PID}}$, and $S_{\text{frag}}$, modulated by the radial moment $r_\lambda$. This interaction is further shaped by a cubic nonlinearity and wrapped in the activation function $\tanh^3$. Although $n_{\text{pf}}$ and $E_Q$ are omitted, the chosen observables capture the dominant patterns relevant for quark–gluon discrimination.

## 6 Outlook

We have shown that quark-gluon taggers trained on low-level jet constituents, despite their complexity, rely on a small set of physically meaningful features. By examining the latent representations of a trained ParticleNet-Lite, we found that much of its performance can be attributed to a few directions, closely related to ($i$) jet multiplicity, ($ii$) radial energy flow, and ($iii$) fragmentation. These directions are not hard-coded into the network but learned, i.e. the training re-discovers the relevant physics and combines it with subtle additional structures.

Beyond confirming the established observables, our analysis suggests new, refined combinations of features that are not immediately obvious from theory. An example is $r_\lambda$ to isolate radial jet structure while remaining decorrelated from multiplicity. Combinations involving fragmentation entropy or charge-related observables show how ParticleNet uses information not captured by the leading substructure observables. We address the question of how to utilize particle identification, a challenge for constituent-based taggers, through its entropy $S_{\text{PID}}$. It quantifies the diversity of particle types and is strongly correlated with one of the leading latent directions. This indicates that the network leverages not just the presence of charged particles, but also the variety of particle types.

Our Shapley value analysis with the SHAP framework highlights both the potential and the limitations of feature attribution in jet tagging. While SHAP successfully identifies physically meaningful observables, its assumption of independent inputs leads to distorted attributions in the presence of correlations. Using decorrelated input features restores consistency with physics expectations. This underscores the importance of careful input preparation when applying SHAP to strongly correlated jet observables.

Finally, we employed symbolic regression to derive simple formulas for these observables that can accurately reproduce the network output. While formulas in terms of only one observable follow the expected pattern, adding a second observable gives us valuable information about additional uncorrelated distinctive power. When allowed to use the seven leading observables, the learned formula only uses five and finds a good compromise between complexity and power. It almost matches the performance of the corresponding trained network.

In the longer term, the compact formulas obtained through symbolic regression could be explored as fast surrogates for full ML taggers in experimental analyses. Their simple ana-

lytic form enables rapid evaluation on large-scale datasets and within environments where computational speed is a critical constraint. While they may not capture the full complexity of a network, such formulas could provide a practical compromise between performance and computational efficiency.

Our set of XAI tools allows us to understand trained precision networks without compromising the training objective or performance. The results show how ML classifiers or other trained networks can inspire theory by highlighting patterns that might otherwise be missed. For networks trained on data, this XAI methodology can connect data-driven methods to physically meaningful information. For applications, understanding the network output and the strong regularization of formulas will allow us to improve resilience and reduce systematic errors.

# Acknowledgements

We would like to thank the Baden-Württemberg-Stiftung for financing through the program *Internationale Spitzenforschung*, project *Uncertainties — Teaching AI its Limits* (BWST_IF2020-010). This work is supported by the Deutsche Forschungsgemeinschaft (DFG, German Research Foundation) under grant 396021762 – TRR 257 *Particle Physics Phenomenology after the Higgs Discovery*. The authors acknowledge support by the state of Baden-Württemberg through bwHPC and the German Research Foundation (DFG) through grant no INST 39/963-1 FUGG (bwForCluster NEMO). SV is supported by the Konrad Zuse School of Excellence in Learning and Intelligent Systems (ELIZA) through the DAAD programme Konrad Zuse Schools of Excellence in Artificial Intelligence, sponsored by the Federal Ministry of Education and Research.

# A  Supplementary tables

| PCA Combination | AUC | $\text{Rej}_{30\%}$ | PCA Combination | AUC | $\text{Rej}_{30\%}$ |
|---|---|---|---|---|---|
| $PC_1$, $PC_2$, $PC_3$, $PC_4$, $PC_5$ | 0.901 | 93.33 | $PC_2$, $PC_3$, $PC_4$, $PC_5$ | 0.774 | 17.36 |
| $PC_1$, $PC_2$, $PC_4$, $PC_5$ | 0.898 | 78.60 | $PC_2$, $PC_4$, $PC_5$ | 0.733 | 10.67 |
| $PC_1$, $PC_2$, $PC_3$, $PC_5$ | 0.898 | 87.84 | $PC_3$, $PC_4$, $PC_5$ | 0.728 | 12.08 |
| $PC_1$, $PC_2$, $PC_5$ | 0.896 | 86.15 | $PC_2$, $PC_3$, $PC_4$ | 0.727 | 9.80 |
| $PC_1$, $PC_2$, $PC_3$, $PC_4$ | 0.893 | 78.60 | $PC_2$, $PC_3$, $PC_5$ | 0.718 | 10.87 |
| $PC_1$, $PC_3$, $PC_4$, $PC_5$ | 0.893 | 81.45 | $PC_2$, $PC_4$ | 0.686 | 7.49 |
| $PC_1$, $PC_2$, $PC_3$ | 0.891 | 74.67 | $PC_2$, $PC_5$ | 0.681 | 7.13 |
| $PC_1$, $PC_4$, $PC_5$ | 0.891 | 73.44 | $PC_3$, $PC_4$ | 0.676 | 7.79 |
| $PC_1$, $PC_2$, $PC_4$ | 0.890 | 75.93 | $PC_2$, $PC_3$ | 0.668 | 7.31 |
| $PC_1$, $PC_3$, $PC_5$ | 0.889 | 78.60 | $PC_4$, $PC_5$ | 0.665 | 7.70 |
| $PC_1$, $PC_2$ | 0.888 | 80.00 | $PC_2$ | 0.634 | 5.57 |
| $PC_1$, $PC_5$ | 0.887 | 72.26 | $PC_3$, $PC_5$ | 0.627 | 7.32 |
| $PC_1$, $PC_3$, $PC_4$ | 0.886 | 75.93 | $PC_4$ | 0.611 | 5.45 |
| $PC_1$, $PC_3$ | 0.883 | 74.67 | $PC_3$ | 0.575 | 5.42 |
| $PC_1$, $PC_4$ | 0.882 | 73.44 | $PC_5$ | 0.549 | 4.18 |
| $PC_1$ | 0.879 | 74.67 | | | |

Table 7: Feedforward classifier trained on different PC combinations using the `Pythia` dataset, showing AUC and 30% quark rejection rate.

| PCA Combination | AUC | $\text{Rej}_{30\%}$ | PCA Combination | AUC | $\text{Rej}_{30\%}$ |
|---|---|---|---|---|---|
| $PC_1$, $PC_2$, $PC_3$, $PC_4$, $PC_5$ | 0.831 | 47.19 | $PC_2$, $PC_3$, $PC_4$, $PC_5$ | 0.715 | 11.49 |
| $PC_1$, $PC_2$, $PC_3$, $PC_5$ | 0.831 | 47.16 | $PC_2$, $PC_3$, $PC_4$ | 0.692 | 9.89 |
| $PC_1$, $PC_2$, $PC_3$, $PC_4$ | 0.831 | 46.19 | $PC_2$, $PC_3$, $PC_5$ | 0.692 | 9.91 |
| $PC_1$, $PC_2$, $PC_4$, $PC_5$ | 0.831 | 44.36 | $PC_2$, $PC_3$ | 0.667 | 8.89 |
| $PC_1$, $PC_2$, $PC_3$ | 0.831 | 45.71 | $PC_2$, $PC_4$, $PC_5$ | 0.654 | 7.87 |
| $PC_1$, $PC_2$, $PC_4$ | 0.830 | 44.36 | $PC_2$, $PC_4$ | 0.646 | 7.47 |
| $PC_1$, $PC_2$, $PC_5$ | 0.830 | 45.71 | $PC_3$, $PC_4$, $PC_5$ | 0.637 | 6.54 |
| $PC_1$, $PC_2$ | 0.830 | 45.71 | $PC_3$, $PC_5$ | 0.629 | 6.25 |
| $PC_1$, $PC_3$, $PC_4$, $PC_5$ | 0.814 | 43.92 | $PC_3$, $PC_4$ | 0.620 | 5.93 |
| $PC_1$, $PC_3$, $PC_4$ | 0.814 | 45.71 | $PC_2$, $PC_5$ | 0.615 | 6.35 |
| $PC_1$, $PC_3$, $PC_5$ | 0.814 | 44.80 | $PC_3$ | 0.611 | 5.43 |
| $PC_1$, $PC_3$ | 0.813 | 44.36 | $PC_2$ | 0.609 | 6.22 |
| $PC_1$, $PC_4$, $PC_5$ | 0.812 | 49.23 | $PC_4$, $PC_5$ | 0.553 | 4.44 |
| $PC_1$, $PC_5$ | 0.811 | 47.66 | $PC_4$ | 0.537 | 3.76 |
| $PC_1$, $PC_4$ | 0.811 | 48.17 | $PC_5$ | 0.530 | 3.90 |
| $PC_1$ | 0.811 | 47.66 | | | |

Table 8: Feedforward classifier trained on different PC combinations using the `Herwig` dataset, showing AUC and 30% quark rejection rate

| PCA Combination | AUC | Rej$_{30\%}$ | PCA Combination | AUC | Rej$_{30\%}$ |
|---|---|---|---|---|---|
| PC$_1$, PC$_2$, PC$_3$, PC$_4$, PC$_5$ | 0.829 | 41.87 | PC$_3$, PC$_4$ | 0.805 | 43.08 |
| PC$_1$, PC$_2$, PC$_3$, PC$_4$ | 0.828 | 43.08 | PC$_1$, PC$_2$, PC$_4$ | 0.805 | 23.33 |
| PC$_1$, PC$_2$, PC$_3$, PC$_5$ | 0.828 | 38.62 | PC$_1$, PC$_2$ | 0.803 | 22.97 |
| PC$_2$, PC$_3$, PC$_4$, PC$_5$ | 0.828 | 43.08 | PC$_2$, PC$_4$ | 0.803 | 22.40 |
| PC$_2$, PC$_3$, PC$_5$ | 0.828 | 40.00 | PC$_2$ | 0.801 | 21.85 |
| PC$_1$, PC$_2$, PC$_3$ | 0.827 | 39.65 | PC$_3$, PC$_5$ | 0.801 | 34.20 |
| PC$_2$, PC$_3$, PC$_4$ | 0.827 | 41.87 | PC$_1$, PC$_3$ | 0.783 | 32.46 |
| PC$_2$, PC$_3$ | 0.826 | 39.30 | PC$_3$ | 0.780 | 32.70 |
| PC$_1$, PC$_3$, PC$_4$, PC$_5$ | 0.816 | 42.26 | PC$_1$, PC$_4$, PC$_5$ | 0.767 | 22.51 |
| PC$_3$, PC$_4$, PC$_5$ | 0.816 | 42.67 | PC$_4$, PC$_5$ | 0.764 | 20.55 |
| PC$_1$, PC$_2$, PC$_4$, PC$_5$ | 0.816 | 33.43 | PC$_1$, PC$_4$ | 0.759 | 17.78 |
| PC$_1$, PC$_3$, PC$_4$ | 0.813 | 42.67 | PC$_4$ | 0.756 | 17.85 |
| PC$_1$, PC$_2$, PC$_5$ | 0.811 | 27.48 | PC$_1$, PC$_5$ | 0.679 | 9.45 |
| PC$_2$, PC$_4$, PC$_5$ | 0.810 | 28.00 | PC$_1$ | 0.641 | 7.09 |
| PC$_2$, PC$_5$ | 0.809 | 24.75 | PC$_5$ | 0.563 | 4.93 |
| PC$_1$, PC$_3$, PC$_5$ | 0.807 | 36.72 | | | |

Table 9: Feedforward classifier trained on different PC combinations using the `Pythia` PCA directions applied to the `Herwig` dataset, showing AUC and 30% quark rejection rate

| complexity | equation | loss | AUC | Rej$_{30\%}$ | $\Delta C$ |
|---|---|---|---|---|---|
| 1 | 0.5 | 0.084 | 0.5 | 3.33 | - |
| 3 | $1.6 - S_{\text{PID}}$ | 0.0054 | 0.842 | 61.84 | 6.69 |
| 5 | $\tanh\left(7.1 \cdot (1 - 0.52 \cdot S_{\text{PID}})^3\right)$ | 0.00065 | 0.842 | 61.84 | 2.79 |
| 6 | $\tanh^2\left(\frac{1.14}{S_{\text{PID}}^3}\right)$ | 0.00028 | 0.842 | 61.84 | 1.74 |
| 7 | $\tanh\left(2.86 \cdot \left(0.48 - \frac{1}{S_{\text{PID}}}\right)^2\right)$ | 0.00021 | 0.842 | 61.84 | 1.93 |
| 8 | $0.96 \cdot \tanh^2\left(\frac{1.18}{S_{\text{PID}}^3}\right)$ | 0.00006 | 0.842 | 61.84 | 2.23 |
| 9 | $0.96 \cdot \tanh\left(2.46 \cdot \left(-0.30 + \frac{1}{S_{\text{PID}}}\right)^3\right)$ | 0.00004 | 0.842 | 61.84 | 2.38 |

Table 10: 1D Symbolic regression tables for $S_{\text{PID}}$

| complexity | equation | loss | AUC | Rej$_{30\%}$ | $\Delta C$ |
|---|---|---|---|---|---|
| 1 | 0.5 | 0.014 | 0.5 | 3.33 | - |
| 4 | $4.3 \cdot \left(\frac{1}{r_\lambda}\right)^{0.5}$ | 0.003 | 0.637 | 6.46 | 7.39 |
| 5 | $0.78 - 0.0032 \cdot r_\lambda$ | 0.0003 | 0.637 | 6.46 | 4.52 |
| 6 | $0.85 \cdot (1.0 - 0.0019 \cdot r_\lambda)^3$ | 0.0001 | 0.637 | 6.46 | 3.00 |
| 7 | $(0.92 - \tanh(0.002 \cdot r_\lambda))^2$ | 0.00011 | 0.637 | 6.46 | 2.80 |
| 8 | $\tanh\left(\frac{114.82}{r_\lambda + 36.44}\right) - 0.23$ | 0.00009 | 0.637 | 6.46 | 2.31 |
| 9 | $0.78 \cdot \left((1.0 - 0.0051 \cdot r_\lambda)^2 + 0.099\right)^{0.5}$ | 0.00005 | 0.637 | 6.46 | 2.03 |
| 10 | $\left((0.59 - \tanh(0.0038 \cdot r_\lambda))^2\right)^{0.5} + 0.22$ | 0.00005 | 0.637 | 6.46 | 1.81 |

Table 11: 1D Symbolic regression tables for $r_\lambda$

| complexity | equation | loss | AUC | $\text{Rej}_{30\%}$ | $\Delta C$ |
|---|---|---|---|---|---|
| 1 | $p_T D$ | 0.058 | 0.807 | 26.87 | 21.19 |
| 2 | $p_T D^{0.5}$ | 0.038 | 0.807 | 26.87 | 15.52 |
| 3 | $1.6 \cdot p_T D$ | 0.015 | 0.807 | 26.87 | 10.78 |
| 5 | $\tanh\left(17.17 \cdot p_T D^3\right)$ | 0.0009 | 0.807 | 26.87 | 2.15 |
| 6 | $\tanh^9\left(5.25 \cdot p_T D\right)$ | 0.0005 | 0.807 | 26.87 | 1.81 |
| 7 | $0.92 \cdot \tanh\left(19.49 \cdot p_T D^3\right)$ | 0.00014 | 0.807 | 26.87 | 1.04 |
| 9 | $0.92 \cdot \tanh\left(22.1 \cdot (p_T D - 0.01)^3\right)$ | 0.00011 | 0.807 | 26.87 | 0.87 |
| 10 | $\tanh^2\left(21.33 \cdot p_T D^3 + 0.35\right) - 0.087$ | 0.00010 | 0.807 | 26.87 | 0.80 |

Table 12: 1D Symbolic regression tables for $p_T D$

| complexity | equation | loss | AUC | $\text{Rej}_{30\%}$ | $\Delta C$ |
|---|---|---|---|---|---|
| 1 | $0.5$ | 0.076 | 0.5 | 3.33 | – |
| 3 | $0.97 - C_{0.2}$ | 0.033 | 0.793 | 58.41 | 15.35 |
| 4 | $2.0 \cdot (1.0 - 0.79 \cdot C_{0.2})^3$ | 0.012 | 0.793 | 58.41 | 10.26 |
| 5 | $2.8 \cdot (0.77 \cdot C_{0.2} - 1.0)^4$ | 0.010 | 0.793 | 58.41 | 8.09 |
| 6 | $\tanh\left(19.66 \cdot (1 - 0.72 \cdot C_{0.2})^9\right)$ | 0.0042 | 0.793 | 58.41 | 5.78 |
| 7 | $\tanh\left(\frac{0.026}{(0.23 - C_{0.2})^2}\right)$ | 0.0014 | 0.793 | 58.41 | 4.28 |
| 9 | $\tanh\left(343.13 \cdot (0.72 \cdot C_{0.2} - 1)^{18} + 0.22\right)$ | 0.00040 | 0.793 | 58.41 | 1.54 |
| 10 | $\tanh\left(88.98 \cdot \sqrt{(1 - 0.72 \cdot C_{0.2})^{27} + 6.72 \cdot 10^{-6}}\right)$ | 0.00038 | 0.793 | 58.41 | 1.72 |

Table 13: 1D Symbolic regression tables for $C_{0.2}$

| complexity | equation | loss | AUC | $\text{Rej}_{30\%}$ | $\Delta C$ |
|---|---|---|---|---|---|
| 1 | $0.49$ | 0.01183 | 0.5 | 3.33 | - |
| 5 | $0.43 \cdot \left(\frac{1}{E_Q}\right)^{0.25}$ | 0.01108 | 0.483 | 4.15 | 10.45 |
| 6 | $E_Q^3 - E_Q + 0.82$ | 0.00206 | 0.617 | 7.37 | 8.39 |
| 7 | $E_Q^4 - E_Q + 0.9$ | 0.00060 | 0.621 | 7.66 | 6.01 |
| 8 | $\tanh\left(E_Q^3 + 1.75 \cdot (1 - 0.83 \cdot E_Q)^3\right)$ | 0.00046 | 0.621 | 7.66 | 6.01 |
| 9 | $1.2 \cdot E_Q^4 - 1.2 \cdot E_Q + 0.99$ | 0.00017 | 0.62 | 7.66 | 2.17 |
| 10 | $(E_Q - 0.031)^3 + \tanh\left(1.76 \cdot (1 - 0.84 \cdot E_Q)^3\right)$ | 0.00005 | 0.62 | 7.65 | 2.57 |

Table 14: 1D Symbolic regression tables for $E_Q$

| complexity | equation | loss | AUC | $\text{Rej}_{30\%}$ | $\Delta C$ |
|---|---|---|---|---|---|
| 1 | $0.5$ | 0.081 | 0.5 | 3.33 | 7.34 |
| 4 | $-\tanh\left(S_{\text{frag}} - 3.5\right)$ | 0.017 | 0.928 | 38.97 | 3.59 |
| 5 | $-\tanh^3\left(S_{\text{frag}} - 3.9\right)$ | 0.0027 | 0.928 | 38.97 | 3.59 |
| 6 | $\tanh^2\left(\frac{18.08}{S_{\text{frag}}^3}\right)$ | 0.0005 | 0.928 | 38.97 | 1.80 |
| 7 | $\tanh\left(19.94 \cdot \left(0.2 - \frac{1}{S_{\text{frag}}}\right)^2\right)$ | 0.0005 | 0.928 | 38.97 | 1.50 |
| 8 | $\left(\tanh\left(\frac{19.38}{S_{\text{frag}}^3}\right) - 0.026\right)^2$ | 0.00015 | 0.928 | 38.97 | 3.02 |
| 10 | $\tanh\left(\frac{5.18}{1.71 \cdot (0.87 \cdot S_{\text{frag}} - 1)^4 + 2.8}\right)$ | 0.00009 | 0.928 | 38.88 | 0.93 |

Table 15: 1D Symbolic regression tables for $S_{\text{frag}}$

| complexity | equation | loss | AUC | Rej$_{30\%}$ | $\Delta C$ |
|---|---|---|---|---|---|
| 1 | $0.51$ | 0.08625 | 0.50 | 3.33 | - |
| 3 | $1.6 - S_{\text{PID}}$ | 0.00832 | 0.842 | 61.84 | 6.71 |
| 4 | $1.7 \cdot \left(1.0 - 0.84 \cdot n_{\text{pf}}\right)^3$ | 0.00656 | 0.840 | 52.32 | 4.00 |
| 5 | $\tanh\left(7.16 \cdot \left(1 - 0.52 \cdot S_{\text{PID}}\right)^3\right)$ | 0.00330 | 0.842 | 61.86 | 2.89 |
| 6 | $\tanh^2\left(\frac{1.15}{S_{\text{PID}}^3}\right)$ | 0.00279 | 0.847 | 61.84 | 2.12 |
| 7 | $\tanh\left(\frac{1.13 - n_{\text{pf}}}{S_{\text{PID}}^3}\right)$ | 0.00047 | 0.847 | 60.99 | 2.48 |
| 8 | $\tanh\left(\frac{1.34 \cdot \left(1 - 0.82 \cdot n_{\text{pf}}\right)^{3/2}}{S_{\text{PID}}^3}\right)$ | 0.00037 | 0.847 | 60.31 | 1.41 |
| 9 | $\tanh\left(\frac{n_{\text{pf}} - 1.06}{\left(0.04 - S_{\text{PID}}\right)^3}\right)$ | 0.00035 | 0.847 | 60.53 | 1.59 |
| 10 | $0.97 \cdot \tanh\left(\frac{1.38 \cdot \left(1 - 0.81 \cdot n_{\text{pf}}\right)^{3/2}}{S_{\text{PID}}^3}\right)$ | 0.00020 | 0.847 | 60.31 | 1.68 |
| 11 | $0.94 \cdot \tanh^{0.5}\left(\frac{1.23 \cdot \left(0.90 \cdot n_{\text{pf}} - 1\right)^2}{S_{\text{PID}}^6}\right)$ | 0.00019 | 0.847 | 60.96 | 1.25 |
| 12 | $0.95 \cdot \tanh\left(\left(0.19 + \frac{1.10 - n_{\text{pf}}}{S_{\text{PID}}^2}\right)^2\right)$ | 0.00012 | 0.847 | 59.95 | 1.28 |
| 13 | $\tanh^{0.5}\left(\left(-0.09 + \frac{n_{\text{pf}} - 1.02}{S_{\text{PID}}^3}\right)^2\right) - 0.057$ | 0.00011 | 0.847 | 60.49 | 1.53 |
| 14 | $-0.081 \cdot S_{\text{PID}} + \tanh\left(\left(-0.45 + \frac{n_{\text{pf}} - 0.89}{S_{\text{PID}}^3}\right)^2\right)$ | 0.00009 | 0.847 | 60.51 | 1.49 |
| 15 | $-0.076 \cdot S_{\text{PID}} + \tanh^{0.5}\left(\left(-0.15 + \frac{n_{\text{pf}} - 0.99}{S_{\text{PID}}^3}\right)^2\right)$ | 0.00009 | 0.847 | 60.96 | 1.33 |
| 16 | $-0.12 \cdot S_{\text{PID}} + \tanh\left(\left(-0.48 + \frac{n_{\text{pf}} - 0.87}{S_{\text{PID}}^3}\right)^2\right) + 0.03$ | 0.00008 | 0.847 | 60.61 | 1.36 |
| 17 | $-0.12 \cdot \tanh\left(S_{\text{PID}}^2\right) + \tanh^{0.5}\left(\left(-0.15 + \frac{n_{\text{pf}} - 1.01}{S_{\text{PID}}^3}\right)^2\right)$ | 0.00008 | 0.847 | 61.16 | 1.27 |
| 19 | $-0.12 \cdot \tanh\left(S_{\text{PID}}^2\right) + \tanh^{0.5}\left(\left(-0.15 + \frac{1.03 \cdot n_{\text{pf}} - 1.03}{S_{\text{PID}}^3}\right)^2\right)$ | 0.00007 | 0.847 | 61.12 | 1.35 |
| 20 | $\tanh^{0.5}\left(\left(-0.15 + \frac{n_{\text{pf}} - 1.02}{S_{\text{PID}}^3}\right)^2\right) - 0.12 \cdot \tanh\left(\left(n_{\text{pf}}^3 + S_{\text{PID}}\right)^2\right)$ | 0.00007 | 0.847 | 60.85 | 1.35 |
| 21 | $-0.12 \cdot \tanh\left(S_{\text{PID}}^2\right) + \tanh^{0.5}\left(\left(\frac{n_{\text{pf}} - 1.02}{n_{\text{pf}}^4 + S_{\text{PID}}^3} - 0.15\right)^2\right)$ | 0.00007 | 0.847 | 61.05 | 1.34 |
| 22 | $-0.12 \cdot \tanh\left(S_{\text{PID}}^2\right) + \tanh^{0.5}\left(\left(\frac{n_{\text{pf}} - 1.01}{n_{\text{pf}}^{9/2} + S_{\text{PID}}^3} - 0.15\right)^2\right)$ | 0.00007 | 0.847 | 61.12 | 1.27 |

Table 16: 2D Symbolic regression tables for $n_{\text{pf}}$ and $S_{\text{PID}}$

| complexity | equation | loss | AUC | $\text{Rej}_{30\%}$ | $\Delta C$ |
|---|---|---|---|---|---|
| 1 | $p_T D$ | 0.07802 | 0.807 | 26.87 | 21.19 |
| 2 | $p_T D^{0.5}$ | 0.05645 | 0.807 | 26.87 | 15.52 |
| 3 | $0.93 - n_{\text{pf}}$ | 0.02289 | 0.840 | 52.32 | 11.05 |
| 4 | $1.7 \cdot \left(1.0 - 0.84 \cdot n_{\text{pf}}\right)^3$ | 0.00592 | 0.840 | 52.32 | 4.13 |
| 5 | $\tanh\left(\frac{0.08}{n_{\text{pf}}^2}\right)$ | 0.00406 | 0.840 | 52.32 | 3.42 |
| 6 | $-\tanh\left(0.22 - \frac{p_T D}{n_{\text{pf}}}\right)$ | 0.00317 | 0.842 | 52.14 | 2.41 |
| 7 | $\tanh\left(\frac{0.26 \cdot p_T D}{n_{\text{pf}}^2}\right)$ | 0.00136 | 0.844 | 55.01 | 1.02 |
| 8 | $\tanh^3\left(\frac{p_T D^2 + 0.32}{n_{\text{pf}}}\right)$ | 0.00111 | 0.844 | 54.00 | 1.48 |
| 9 | $\tanh\left(\left(p_T D - 0.27 + \frac{0.27}{n_{\text{pf}}}\right)^2\right)$ | 0.00093 | 0.845 | 54.70 | 1.39 |
| 10 | $0.94 \cdot \tanh\left(\left(p_T D + \frac{0.068}{n_{\text{pf}}^2}\right)^2\right)$ | 0.00076 | 0.845 | 54.00 | 0.73 |
| 11 | $\tanh^2\left((p_T D + 0.35)^3 + \frac{0.08}{n_{\text{pf}}^2}\right)$ | 0.00070 | 0.845 | 54.17 | 1.18 |
| 12 | $0.95 \cdot \tanh\left(4.65 \cdot p_T D^3 + \frac{0.03}{n_{\text{pf}}^3}\right)$ | 0.00044 | 0.845 | 53.94 | 0.64 |
| 13 | $0.95 \cdot \tanh\left((p_T D + 0.41)^6 + \frac{0.025}{n_{\text{pf}}^3}\right)$ | 0.00044 | 0.845 | 53.88 | 0.94 |
| 14 | $0.95 \cdot \tanh\left(4.41 \cdot p_T D^3 + 0.01 + \frac{0.02}{n_{\text{pf}}^3}\right)$ | 0.00042 | 0.845 | 53.71 | 0.60 |
| 16 | $0.95 \cdot \tanh\left(6.16 \cdot (0.21 - p_T D)^2 + 0.02 \cdot \left(0.49 + \frac{1}{n_{\text{pf}}}\right)^3\right)$ | 0.00037 | 0.846 | 54.23 | 0.58 |
| 17 | $0.95 \cdot \tanh\left(5.28 \cdot p_T D^3 + \frac{0.02}{\left(n_{\text{pf}} - \frac{0.0005}{p_T D^3}\right)^3}\right)$ | 0.00031 | 0.846 | 53.13 | 1.07 |
| 18 | $0.95 \cdot \tanh\left(0.019 \cdot \left(0.41 + \frac{1}{n_{\text{pf}}}\right)^3 + \left(p_T D \cdot (n_{\text{pf}} + 2.30) - 0.63\right)^2\right)$ | 0.00031 | 0.846 | 53.82 | 0.64 |
| 19 | $\tanh\left(5.59 \cdot p_T D^3 + 4836.5 \cdot \left((0.37 - p_T D)^3 + \frac{0.017}{n_{\text{pf}}}\right)^3\right) - 0.047$ | 0.00025 | 0.846 | 53.76 | 0.80 |
| 21 | $0.95 \cdot \tanh\left(5.79 \cdot p_T D^3 + 0.03 \cdot \left(-0.23 + \frac{1}{n_{\text{pf}} + 9.96 \cdot (p_T D - 0.39)^3}\right)^3\right)$ | 0.00022 | 0.846 | 52.63 | 0.68 |

Table 17: 2D Symbolic regression tables for $n_{\text{pf}}$ and $p_T D$

| complexity | equation | loss | AUC | Rej$_{30\%}$ | $\Delta C$ |
|---|---|---|---|---|---|
| 1 | $0.50$ | 0.10417 | 0.5 | 3.33 | - |
| 3 | $0.94 - n_{\text{pf}}$ | 0.03430 | 0.840 | 52.32 | 11.12 |
| 4 | $1.7 \cdot \left(1.0 - 0.84 \cdot n_{\text{pf}}\right)^3$ | 0.01592 | 0.840 | 52.32 | 3.90 |
| 5 | $\tanh\left(\frac{0.03}{n_{\text{pf}}^3}\right)$ | 0.01307 | 0.840 | 52.32 | 2.90 |
| 6 | $\tanh^6\left(\frac{0.57}{n_{\text{pf}}}\right)$ | 0.01262 | 0.840 | 52.32 | 2.05 |
| 7 | $\tanh\left(\frac{0.04}{n_{\text{pf}}^3 + 0.01}\right)$ | 0.01226 | 0.840 | 52.32 | 1.15 |
| 8 | $\tanh\left(\frac{0.31 \cdot \sqrt{\frac{1}{r_\lambda}}}{n_{\text{pf}}^3}\right)$ | 0.00612 | 0.852 | 46.04 | 2.04 |
| 9 | $\tanh^2\left(\frac{1.28 \cdot \sqrt{\frac{1}{r_\lambda}}}{n_{\text{pf}}^2}\right)$ | 0.00470 | 0.852 | 41.25 | 2.37 |
| 10 | $\tanh^9\left(\frac{189.7}{n_{\text{pf}} \cdot (r_\lambda + 198.6)}\right)$ | 0.00391 | 0.855 | 42.84 | 2.06 |
| 11 | $\tanh^3\left(9.27 \cdot \left(\sqrt{\frac{1}{r_\lambda}} + \frac{0.091}{n_{\text{pf}}}\right)^2\right)$ | 0.00337 | 0.856 | 43.52 | 1.90 |
| 12 | $\tanh^9\left(7.32 \cdot \sqrt{\frac{1}{r_\lambda}} + \frac{0.13}{n_{\text{pf}}^2}\right)$ | 0.00238 | 0.858 | 46.00 | 1.47 |
| 13 | $\tanh^9\left(\frac{123.8}{r_\lambda + 68.67} + \frac{0.13}{n_{\text{pf}}^2}\right)$ | 0.00190 | 0.859 | 48.59 | 1.64 |
| 14 | $\tanh^{18}\left(\frac{236.17}{r_\lambda + 120.8} + \frac{0.13}{n_{\text{pf}}^2}\right)$ | 0.00184 | 0.859 | 49.36 | 1.68 |
| 15 | $\left(1.1 - n_{\text{pf}}\right) \cdot \tanh\left(181.1 \cdot \left(\sqrt{\frac{1}{r_\lambda}} + \frac{0.003}{n_{\text{pf}}^3}\right)^3\right)$ | 0.00115 | 0.860 | 51.81 | 1.41 |
| 16 | $\left(1.1 - n_{\text{pf}}\right) \cdot \tanh^3\left(54.53 \cdot \left(\sqrt{\frac{1}{r_\lambda}} + \frac{0.003}{n_{\text{pf}}^3}\right)^2\right)$ | 0.00083 | 0.860 | 51.81 | 1.43 |
| 17 | $\left(1.1 - n_{\text{pf}}\right) \cdot \tanh^{1.5}\left(268.4 \cdot \left(\sqrt{\frac{1}{r_\lambda}} + \frac{0.003}{n_{\text{pf}}^3}\right)^3\right)$ | 0.00082 | 0.860 | 51.81 | 1.77 |
| 18 | $\tanh\left(\frac{0.36}{n_{\text{pf}}}\right) \cdot \tanh\left(464.46 \cdot \left(\sqrt{\frac{1}{r_\lambda}} - 0.05 + \frac{0.01}{n_{\text{pf}}^2}\right)^3\right)$ | 0.00078 | 0.860 | 51.81 | 1.59 |
| 19 | $\tanh\left(\frac{0.36}{n_{\text{pf}}}\right) \cdot \tanh\left(1273.42 \cdot \left(0.01 \cdot \left(-0.60 + \frac{1}{n_{\text{pf}}}\right)^2 + \sqrt{\frac{1}{r_\lambda}}\right)^4\right)$ | 0.00076 | 0.860 | 51.81 | 1.51 |
| 20 | $\tanh\left(\frac{0.36}{n_{\text{pf}}}\right) \cdot \tanh\left(464.25 \cdot \left(\sqrt{\frac{1}{r_\lambda}} - 0.04 + \frac{0.007}{(n_{\text{pf}} - 0.05)^2}\right)^3\right)$ | 0.00075 | 0.860 | 51.81 | 1.52 |
| 21 | $\left(-n_{\text{pf}} - 0.59 \cdot \left(\frac{1}{r_\lambda}\right)^{0.5} + 1.2\right) \cdot \tanh\left(1472.86 \cdot \left(\sqrt{\frac{1}{r_\lambda}} + \frac{0.0026}{n_{\text{pf}}^3}\right)^4\right)$ | 0.00069 | 0.860 | 54.41 | 1.60 |
| 22 | $\left(-n_{\text{pf}} - 0.57 \cdot \left(\frac{1}{r_\lambda}\right)^{0.5} + 1.2\right) \cdot \left(\tanh^3\left(272.2 \cdot \left(\sqrt{\frac{1}{r_\lambda}} + \frac{0.003}{n_{\text{pf}}^3}\right)^3\right)\right)^{0.5}$ | 0.00065 | 0.860 | 54.76 | 1.46 |

Table 18: 2D Symbolic regression tables for $n_{\text{pf}}$ and $r_\lambda$

| complexity | equation | loss | AUC | Rej$_{30\%}$ | $\Delta C$ |
|---|---|---|---|---|---|
| 1 | $0.5$ | 0.08389 | 0.500 | 3.33 | - |
| 3 | $0.93 - n_{\text{pf}}$ | 0.01909 | 0.840 | 52.32 | 11.04 |
| 4 | $1.7 \cdot \left(1.0 - 0.84 \cdot n_{\text{pf}}\right)^3$ | 0.00358 | 0.840 | 52.32 | 4.14 |
| 5 | $\tanh\left(\frac{0.08}{n_{\text{pf}}^2}\right)$ | 0.00156 | 0.840 | 52.32 | 3.53 |
| 6 | $\tanh^6\left(\frac{0.57}{n_{\text{pf}}}\right)$ | 0.00113 | 0.840 | 52.32 | 2.01 |
| 7 | $\tanh\left(\frac{0.04}{n_{\text{pf}}^3 + 0.01}\right)$ | 0.00069 | 0.840 | 52.32 | 1.22 |
| 9 | $0.93 \cdot \tanh\left(\frac{0.055}{\left(n_{\text{pf}} - 0.086\right)^2}\right)$ | 0.00053 | 0.830 | 52.32 | 0.96 |
| 10 | $\tanh\left(\frac{0.04}{n_{\text{pf}}^3 + 0.02 \cdot \sqrt{E_Q}}\right)$ | 0.00047 | 0.840 | 52.36 | 1.29 |
| 11 | $-0.098 \cdot E_Q + \tanh\left(0.11 + \frac{0.03}{n_{\text{pf}}^3}\right)$ | 0.00040 | 0.840 | 52.36 | 0.84 |
| 12 | $0.94 \cdot \tanh\left(\frac{0.06}{\left(n_{\text{pf}} + 0.08 \cdot \sqrt{E_Q}\right)^3}\right)$ | 0.00032 | 0.840 | 52.41 | 0.86 |
| 13 | $0.9 \cdot \tanh\left(\frac{0.04}{\left(n_{\text{pf}} + 0.05 \cdot E_Q\right)^3}\right) + 0.04$ | 0.00026 | 0.840 | 52.69 | 0.88 |
| 14 | $0.92 \cdot \tanh\left(\frac{0.05}{\left(n_{\text{pf}} + 0.06 \cdot \sqrt{E_Q}\right)^3}\right) + 0.027$ | 0.00025 | 0.840 | 52.52 | 0.94 |
| 16 | $0.18 - 0.76 \cdot \tanh\left(0.19 - \frac{0.06}{\left(n_{\text{pf}} + 0.07 \cdot \sqrt{E_Q}\right)^3}\right)$ | 0.00023 | 0.840 | 52.36 | 0.85 |
| 17 | $0.32 - 0.62 \cdot \tanh\left(0.49 - \frac{0.098}{\left(n_{\text{pf}} + 0.13 \cdot \sqrt[4]{E_Q}\right)^3}\right)$ | 0.00022 | 0.840 | 52.52 | 0.87 |
| 18 | $0.42 - 0.52 \cdot \tanh\left(0.92 - \frac{0.24}{\left(n_{\text{pf}} + 0.23 \cdot \sqrt[8]{E_Q}\right)^3}\right)$ | 0.00021 | 0.840 | 52.36 | 0.94 |
| 19 | $0.18 - 0.76 \cdot \tanh\left(0.19 - \frac{0.06}{\left(n_{\text{pf}} + E_Q \cdot \left(0.18 - 0.11 \cdot E_Q\right)\right)^3}\right)$ | 0.00021 | 0.840 | 52.69 | 0.92 |
| 20 | $0.36 - 0.58 \cdot \tanh\left(0.62 - \frac{0.13}{\left(n_{\text{pf}} + 0.32 \cdot \sqrt{E_Q} - 0.17 \cdot E_Q\right)^3}\right)$ | 0.00020 | 0.840 | 52.52 | 0.92 |
| 21 | $0.91 \cdot \tanh\left(\frac{0.05}{\left(n_{\text{pf}} + \frac{0.09 \cdot E_Q}{\sqrt{\left(n_{\text{pf}} - E_Q\right)^2 + 0.58}}\right)^3}\right) + 0.034$ | 0.00017 | 0.840 | 52.08 | 0.99 |
| 22 | $0.91 \cdot \tanh\left(\frac{0.05}{\left(n_{\text{pf}} + \frac{0.07 \cdot E_Q}{\sqrt{\sqrt{\left(n_{\text{pf}} - E_Q\right)^2} + 0.10}}\right)^3}\right) + 0.034$ | 0.00017 | 0.840 | 51.92 | 0.95 |

Table 19: 2D Symbolic regression tables for $n_{\text{pf}}$ and $E_Q$

| complexity | equation | loss | AUC | $\text{Rej}_{30\%}$ | $\Delta C$ |
|---|---|---|---|---|---|
| 1 | $0.49$ | $0.09075$ | $0.5$ | $3.33$ | - |
| 3 | $0.92 - n_{\text{pf}}$ | $0.02469$ | $0.840$ | $52.32$ | $11.35$ |
| 4 | $1.7 \cdot \left(1.0 - 0.84 \cdot n_{\text{pf}}\right)^3$ | $0.00795$ | $0.840$ | $52.32$ | $4.25$ |
| 5 | $\tanh\left(\frac{0.082}{n_{\text{pf}}^2}\right)$ | $0.00612$ | $0.840$ | $52.32$ | $3.59$ |
| 6 | $\tanh^6\left(\frac{0.56}{n_{\text{pf}}}\right)$ | $0.00532$ | $0.840$ | $52.32$ | $2.05$ |
| 7 | $\tanh\left(\frac{0.06}{\left(n_{\text{pf}}+0.10\right)^3}\right)$ | $0.00510$ | $0.840$ | $52.32$ | $1.49$ |
| 8 | $\tanh\left(\frac{0.10}{\left(n_{\text{pf}}+0.26\right)^4}\right)$ | $0.00506$ | $0.840$ | $52.32$ | $1.49$ |
| 9 | $0.94 \cdot \tanh\left(0.02 \cdot \left(0.37 + \frac{1}{n_{\text{pf}}}\right)^3\right)$ | $0.00484$ | $0.840$ | $52.32$ | $1.08$ |
| 10 | $\tanh\left(\frac{0.07}{\left(n_{\text{pf}} - \left(C_{0.2} - 0.58\right)^2\right)^2}\right)$ | $0.00444$ | $0.841$ | $61.58$ | $2.00$ |
| 11 | $\tanh^4\left(\frac{0.46}{-n_{\text{pf}} + \left(C_{0.2} - 0.57\right)^2}\right)$ | $0.00417$ | $0.841$ | $60.46$ | $1.32$ |
| 12 | $-\tanh\left(C_{0.2} - \frac{\sqrt{\left(C_{0.2}-0.46\right)^2} + 0.35}{n_{\text{pf}}}\right)$ | $0.00265$ | $0.846$ | $62.81$ | $2.43$ |
| 13 | $\tanh^2\left(C_{0.2} - \frac{\sqrt{\left(C_{0.2}-0.447\right)^2} + 0.48}{n_{\text{pf}}}\right)$ | $0.00233$ | $0.846$ | $61.5$ | $1.55$ |
| 14 | $\tanh^3\left(-C_{0.2} + \sqrt[4]{\left(C_{0.2} - 0.45\right)^2} + \frac{0.53}{n_{\text{pf}}}\right)$ | $0.00199$ | $0.846$ | $59.03$ | $1.34$ |
| 15 | $\tanh^2\left(-\sqrt{C_{0.2}} + \sqrt[4]{\left(C_{0.2} - 0.45\right)^2} + \frac{0.53}{n_{\text{pf}}}\right)$ | $0.00181$ | $0.846$ | $59.03$ | $1.36$ |
| 16 | $\tanh^6\left(131.67 \cdot \left(\left(C_{0.2} - 0.68\right)^2 - 0.05\right)^2 + \frac{0.52}{n_{\text{pf}}}\right)$ | $0.00139$ | $0.847$ | $64.02$ | $1.88$ |
| 17 | $\tanh^3\left(-C_{0.2} + 3.88 \cdot \sqrt{\left(\left(0.71 \cdot C_{0.2} - 1\right)^4 - 0.21\right)^2} + \frac{0.54}{n_{\text{pf}}}\right)$ | $0.00122$ | $0.848$ | $62.34$ | $1.61$ |
| 18 | $\left(0.03 - \tanh^3\left(\frac{82.47 \cdot \left(\left(C_{0.2}-0.68\right)^2 - 0.04\right)^2 + 0.54}{n_{\text{pf}}}\right)\right)^2$ | $0.00089$ | $0.848$ | $65.27$ | $1.30$ |
| 19 | $\left(\tanh\left(-C_{0.2} + 287 \cdot \left(\left(0.70 \cdot C_{0.2} - 1\right)^8 - 0.05\right)^2 + \frac{0.58}{n_{\text{pf}}}\right) - 0.019\right)^3$ | $0.00078$ | $0.848$ | $65.10$ | $1.51$ |
| 21 | $\tanh^3\left(-C_{0.2} + 288.3 \cdot \left(\left(0.70 \cdot C_{0.2} - 1\right)^8 - 0.05\right)^2 + \frac{0.64}{n_{\text{pf}}+0.03}\right) - 0.05$ | $0.00074$ | $0.848$ | $65.19$ | $1.33$ |
| 22 | $\left(\tanh^3\left(-C_{0.2} + 300.9 \cdot \left(\left(0.70 \cdot C_{0.2} - 1\right)^8 - 0.05\right)^2 + \frac{0.89}{n_{\text{pf}}+0.09}\right) - 0.027\right)^2$ | $0.00073$ | $0.848$ | $64.77$ | $1.18$ |

Table 20: 2D Symbolic regression tables for $n_{\text{pf}}$ and $C_{0.2}$

| complexity | equation | | loss | AUC | Rej$_{30\%}$ | $\Delta C$ |
|---|---|---|---|---|---|---|
| 1 | $0.5$ | | 0.09448 | 0.5 | 3.33 | |
| 3 | $0.93 - n_{\text{pf}}$ | | 0.02494 | 0.840 | 52.32 | 11.07 |
| 4 | $1.7 \cdot \left(1.0 - 0.84 \cdot n_{\text{pf}}\right)^3$ | | 0.00669 | 0.840 | 52.32 | 4.06 |
| 5 | $\tanh\left(\dfrac{0.03}{n_{\text{pf}}^3}\right)$ | | 0.00372 | 0.840 | 52.32 | 2.78 |
| 6 | $\tanh^6\left(\dfrac{0.57}{n_{\text{pf}}}\right)$ | | 0.00327 | 0.840 | 52.32 | 2.06 |
| 7 | $\tanh^3\left(\dfrac{1.18}{n_{\text{pf}} \cdot S_{\text{frag}}}\right)$ | | 0.00151 | 0.844 | 54.35 | 1.74 |
| 9 | $0.94 \cdot \tanh\left(\dfrac{0.69}{n_{\text{pf}}^2 \cdot S_{\text{frag}}^2}\right)$ | | 0.00109 | 0.844 | 54.29 | 1.32 |
| 11 | $0.95 \cdot \tanh^2\left(\dfrac{0.73}{S_{\text{frag}} \cdot (n_{\text{pf}} - 0.11)}\right)$ | | 0.00099 | 0.844 | 55.80 | 1.03 |
| 12 | $0.93 \cdot \tanh\left(1.89 \cdot \left(-\dfrac{1}{S_{\text{frag}}} - \dfrac{0.15}{n_{\text{pf}}}\right)^4\right)$ | | 0.00096 | 0.844 | 55.80 | 1.33 |
| 13 | $0.94 \cdot \tanh^2\left(\dfrac{0.57}{(n_{\text{pf}} - 0.13) \cdot (n_{\text{pf}} - S_{\text{frag}})}\right)$ | | 0.00092 | 0.844 | 55.74 | 0.93 |
| 14 | $0.94 \cdot \tanh\left(0.96 \cdot \left(-0.06 - \dfrac{1}{S_{\text{frag}}} - \dfrac{0.18}{n_{\text{pf}}}\right)^4\right)$ | | 0.00089 | 0.844 | 55.31 | 0.95 |
| 15 | $0.95 \cdot \tanh^2\left(0.28 \cdot \left(-0.52 - \dfrac{1}{S_{\text{frag}}} - \dfrac{0.18}{n_{\text{pf}}}\right)^4\right)$ | | 0.00086 | 0.844 | 55.37 | 0.94 |
| 16 | $0.93 \cdot \tanh\left(\dfrac{\left((n_{\text{pf}} - 0.17 \cdot S_{\text{frag}})^2 + \frac{0.81}{S_{\text{frag}}}\right)^2}{n_{\text{pf}}^2}\right)$ | | 0.00082 | 0.844 | 55.31 | 1.60 |
| 17 | $0.94 \cdot \tanh\left(\dfrac{\left((-n_{\text{pf}} + 0.059 \cdot S_{\text{frag}}^2)^2 + \frac{0.81}{S_{\text{frag}}}\right)^2}{n_{\text{pf}}^2}\right)$ | | 0.00045 | 0.845 | 54.53 | 1.11 |
| 18 | $0.94 \cdot \tanh\left(\dfrac{\left(\sqrt{\left(-n_{\text{pf}} + 0.06 \cdot S_{\text{frag}}^2\right)^2 + 0.77}\right)^2}{n_{\text{pf}}^2 \cdot S_{\text{frag}}^2}\right)$ | | 0.00040 | 0.845 | 53.59 | 1.15 |
| 19 | $0.94 \cdot \tanh\left(\dfrac{4 \cdot \left((n_{\text{pf}} - 0.06 \cdot S_{\text{frag}}^2)^2 + \frac{0.40}{S_{\text{frag}}}\right)^2}{n_{\text{pf}}^2}\right)$ | | 0.00030 | 0.845 | 53.88 | 1.09 |
| 21 | $0.93 \cdot \tanh\left(\dfrac{3.90 \cdot \left((-n_{\text{pf}} + 0.06 \cdot S_{\text{frag}}^2)^2 + \frac{0.40}{S_{\text{frag}}}\right)^2}{n_{\text{pf}}^2}\right) + 0.0092$ | | 0.00029 | 0.845 | 53.88 | 0.89 |
| 22 | $0.94 \cdot \tanh^2\left(2.74 \cdot \left(0.19 + \dfrac{\left(-n_{\text{pf}} + 0.05 \cdot S_{\text{frag}}^2\right)^2 + \frac{0.40}{S_{\text{frag}}}}{n_{\text{pf}}}\right)^2\right)$ | | 0.00027 | 0.845 | 53.88 | 1.03 |

Table 21: 2D Symbolic regression tables for $n_{\text{pf}}$ and $S_{\text{frag}}$

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
