# Peer review of "How to Deep-Learn the Theory behind Quark-Gluon Tagging"

_SciPost Physics_

## Round 1 · Referee Report · Anonymous (Referee 1) · 2025-9-30

Report

The manuscript "How to Deep-Learn the Theory behind Quark-Gluon Tagging" studies the problem of discrimination of quark- from gluon-initiated jets in the context of machine learning. The goal of this study is to develop high-level, interpretable observables that have (almost) the same performance as state-of-the-art architectures. The authors approach this problem through various methods for reducing the dimensionality of the relevant observable space, as well as comparing to machine-learned functions of various observables known to perform well at this problem. While this work is very topical and in its motivation and goals is rather novel, due to serious shortcomings in the analysis, I cannot recommend it for publication.

I have many particular points that I will address below, but I will first describe the overall problem I see with this endeavor as laid forth in this paper. Unfortunately, these issues are so pervasive, I do not think that any form of this paper could be publishable without an entirely new analysis.

The goal of this paper, as the authors state several times, is to produce performant and interpretable observables through various machine learning techniques. The performance aspect of this goal is rather well-defined, as a quantifiable metric for discrimination of quark versus gluon jets. The authors mostly focus on the area under the ROC curve (AUC), which may have some problems, but is nevertheless concrete and quantifiable. By contrast, the "interpretability" aspect of the goal is exactly the opposite of this, very heuristic, hand-wavy, and imprecise. In particular, the arguments that the authors make for interpretability are extremely dated, which is very unfortunate.

In particular, I feel like the authors have relied too much on heuristics of QCD, and not concrete calculations, numerical predictions, systematically-improvable approximations, etc., that make QCD the robust theory it is. The interpretations of quark and gluon jets and observables that are useful for discrimination read like that part of the paper was written in 2010. Specifically, in jet physics there has been significant advances in calculational ability, level of precision, and sophistication of techniques over the past 15 years. As a concrete example, in the context of the symbolic regression study in Section 5, the authors train a network to output an appropriately optimal form of observables, given particular input high-level observables, mathematical functions and operations. Again, this approach is rather novel and could be very informative and intriguing.

However, the set of observables and the mathematical functions that the machine can manipulate frankly don't make much sense. First, the high-level observables are a rather ad-hoc set of things that work (more on this later), but with no guiding principle for their inclusion. Second, a particular function that apparently does much of the heavy lifting of the regression is the hyperbolic tangent. I must admit, I can't think of many (or perhaps any?) results from calculations in QCD in which a hyperbolic tangent appears. As such, I have no interpretation for how such a function would arise in QCD, and correspondingly don't understand what the result is supposed to mean.

The authors could have included knowledge from QCD in parallel with the machine learning studies to find a useful and interpretable observable through symbolic regression. For example, for many of the high-level observables considered by the authors, the functional form of the distribution of those observables is known. Further, in many cases, these functional forms are known at a higher accuracy in perturbation theory than the parton shower event simulation employed in this work. The authors could have incorporated this QCD domain knowledge into the regression task, which would be significantly more enlightening as to where these functions could come from. As just three examples:

1) The distribution of particle multiplicity in a jet is well-known to closely follow KNO scaling [1,2]. Further, it is also well-known that the negative binomial distribution describes the multiplicity distribution well, see, e.g., [3].

2) The observable S_PID which measures the entropy of particle identification, is closely related to the jet charge [4]. Dominantly, pions are produced in a jet, and the three pions have distinct electric charges. The distribution of the jet charge is known to be a Gaussian conditioned on multiplicity, by the central limit theorem [5].

3) At leading-logarithmic accuracy, the distribution of IRC safe observables like the energy correlation functions or the jet width take the Sudakov form of an exponential function with a double logarithmic argument.

In each of these examples, the particular function that appears in the distribution has a physical origin, meaning and interpretation. The hyperbolic tangent used in this paper lacks that sort of richness and connection to the physics of QCD. Is there a reason that none of this information is used?

More specific issues follow. I want to emphasize that this is not exhaustive, as well, as sometimes I just identified the first time something was used or stated.

1) In the introduction, the authors mention some subtleties of quark versus gluon tagging, such as ambiguity beyond leading-order. However, in Sec. 2, without any discussion, they apparently use "Monte Carlo" definitions of quark and gluon jets, as the output of a parton shower event simulator, according to selected initiating process. This is not satisfactory, especially given significant work in defining quark and gluon jets, or on jet flavor definitions, etc., over the past decade or more. The authors are welcome to define quarks and gluons in this Monte Carlo way, but their results are then not robust nor do they have any physical meaning.

The approach should instead be opposite. The authors need to provide a robust definition of quark and gluon jets in some, physical, way. E.g., in a semi-supervised or even unlabeled event context, like different populations of different jet types at different pseudorapidities. Then, they could establish the features that a machine learns and exploits in the different populations to classify them as "quark" and "gluon". As such currently, this approach only teaches readers about what Pythia does, which is not Nature.

2) Why the observables in Eq. 8? There is a strange disconnect between the structure of ParticleNet, which is some general, extremely expressive architecture that can capture subtle and non-trivial correlations between particles (or with its Lite variant). Then the authors pick 4 high-level observables that, while each are known to perform decently on the problem of quark versus gluon jets, there is no sense that their combination is "better". Instead, one should approach the comparison in an analogous way as ParticleNet, with some sufficiently expressive high-level observables, that are, possibly IRC safe for robustness and calculability. Such a set of observables would be Energy Flow Polynomials, for example, but many such things could be used. With such a set of high-level observables, then their combination and improvement in performance as the number of observables is increased has some meaning, because more observables enables probing correlations on a smaller scale. Specifically, Energy Flow Polynomials are linear combinations of orthogonal harmonics on n-body phase space.

3) If I understand correctly, the correlations presented in Fig 4 are linear correlations (Pearson correlations). The authors actually do not define what they mean by "correlation" here, unless I missed it. This correlation is woefully insufficient for drawing conclusions. Non-linear correlations between these observables will be significant, so a much better measure of correlation must be employed. There are many measures of non-functional correlations that exist (mutual information, etc). Actually, mutual information is mentioned later, in the study of Shapley values, but still isn't used there.

4) The linear relationships expressed in Eq. 10 or Eq. 13 have no physical meaning. Or, rather, if they do, the authors need to mention that clearly and describe to the reader what that meaning is.

5) For all their statements about learning from the machine, the authors try to do a lot of physical interpretation themselves, which is heuristic at best, and very outdated in its approach. Heuristics like discussed below Eq. 15 for motivating this particular observable were common in the field 15 years ago, but with significant theoretical advances in calculations for problems like this, are not really employed anymore. Anyway, I thought the point of this paper was to go beyond these heuristics anyway.

6) Fig 5: I don't really understand what the caption is saying. Both figures show the correlation coefficient with PC1, PC2 and PC3.

7) Again, what is the reader supposed to learn from Eq. 18? The authors need to spell it out clearly.

8) I don't follow Eq. 20. You want to identify non-linear correlations, so you add a term in the loss function with the covariance. The covariance only encodes linear correlations. So how does this solve the problem? Can't this term vanish for observables that are perfectly correlated, but in a non-linear way?

9) I don't understand the blue-red gradient in Figure 8. What does "Feature Value" mean, and are "High" and "Low" quantifiable?

10) The authors make many, many statements without any quantifiable evidence. For example, the middle paragraph on page 12, in which they interpret what is happening with wpf, is imprecise. Also, in the last full paragraph on page 12, the authors state that these observables are correlated, but provide no evidence of non-linear correlation, like mutual information. How are these observables correlated, quantifiably?

11) With all of the additional interpretation and decorrelation that the authors seem to need to do, I fail to see the utility of Shapley values to this problem. The authors are only applying this analysis to collections of 6 observables, and they find odd results. So, they then work to reduce correlations between observables by hand. How could this ever generalize to much higher dimensionality?

12) Page 14 in "Setup and method": the authors state that they select observables based on "performance and interpretability". What is "Interpretability"? Is this quantified?

13) I guess i don't understand what Delta C in Eq. 21 means. Can the authors provide more discussion about this? It is a bit troubling that seems to explicitly depends on binning. Is this true? I am especially confused through Fig 10 and Table 3. All of these equations listed in Table 3 are monotonically related. As such, the discrimination performance is unchanged from simply measuring npf alone. (I also don't understand the need for Ref 84 for this point; this is a consequence of the original Neyman-Pearson proof.) How can the "calibration" Delta C change? In all of these cases, you just measure npf, and then put that value into a formula. If anything, the expression at complexity 9, which has low Delta C, is impossibly uninterpretable.

14) The expressions in Table 4 are not interpretable in any colloquial sense. I learn no physics by staring at these equations, and the particular forms are highly specialized to the mathematical functions that are allowed. Table 5 and Equation 31 are even less interpretable. The authors should provide a quantifiable "interpretation" of this equation. On page 19 they say that Eq. 31 is interpretable, but provide no interpretation. Why not? How can it be interpreted? The authors present no theoretical calculations of observables in perturbation theory (or even in simplified models of QCD), which I would think would be the minimal baseline for true interpretability.

15) In conclusions the authors state: "Beyond confirming the established observables, our analysis suggests new, refined combinations of features that are not immediately obvious from theory." What does this mean? The authors provide no theory analysis, so the reader has nothing to compare to. QCD theory is not heuristics.

16) I must admit, I don't know what the point of the appendices is. What is the reader supposed to learn from them?

References:

[1] A. M. Polyakov, A Similarity hypothesis in the strong interactions. 1. Multiple hadron production in e+ e- annihilation, Zh. Eksp. Teor. Fiz. 59 (1970) 542–552.

[2] Z. Koba, H. B. Nielsen, and P. Olesen, Scaling of multiplicity distributions in high-energy hadron collisions, Nucl. Phys. B 40 (1972) 317–334.

[3] P. Carruthers and C. C. Shih, Correlations and Fluctuations in Hadronic Multiplicity Distributions: The Meaning of KNO Scaling, Phys. Lett. B 127 (1983) 242–250.

[4] R. D. Field and R. P. Feynman, A Parametrization of the Properties of Quark Jets, Nucl. Phys. B 136 (1978) 1.

[5] Z.-B. Kang, A. J. Larkoski, and J. Yang, Towards a Nonperturbative Formulation of the Jet Charge, Phys. Rev. Lett. 130 (2023), no. 15 151901, [arXiv:2301.09649]

Recommendation

Reject

  • validity: low
  • significance: poor
  • originality: low
  • clarity: poor
  • formatting: reasonable
  • grammar: good

Author:  Ramon Winterhalder  on 2025-12-23  [id 6184]

(in reply to Report 1 on 2025-09-30)
Category:
answer to question
reply to objection

We thank the referee for their careful reading and for raising broader concerns about our goals and methodology. We want to clarify an apparent misunderstanding: our aim is not to derive new QCD predictions or to replace theoretical calculations. Instead, the objective of this work is to develop and test explainable machine learning methods that reveal which features a state-of-the-art tagger encodes.

The symbolic regression and dimensionality-reduction steps are not intended as new perturbative QCD results. Instead, they serve as surrogate models that approximate the network's decision boundary in a compact, human-readable form. The interpretability then lies in the fact that these surrogates consistently align with well-known QCD-motivated observables (multiplicity, radial jet structure, fragmentation hardness, particle-type diversity). In other words, QCD provides the physics interpretation; our methods identify the combinations that the network actually exploits.

To avoid any ambiguity, we now emphasize in the introduction and conclusions that: - Our contribution is methodological: applying XAI techniques to jet tagging, not proposing new analytic forms for QCD distributions. - Interpretability is defined operationally: the extent to which network-encoded features can be mapped onto known physics concepts. - QCD theory and our surrogates are complementary: theory predicts the distributions, while our surrogates uncover the feature usage of ML classifiers.

We believe this clarification addresses the referee's concern that our arguments were "heuristic" or "dated." The heuristics are not meant as a substitute for theory but as the natural language in which physicists interpret which observables a network encodes. Hence, Symbolic Regression was trained on the taggers' predictions. Which means we fit an ML-function. The label of a quark corresponding to a "1" and gluons to a "0" can not be well motivated by QCD laws, so we need functions like the hyperbolic tangent to describe this. We provide more detailed answers to all points below:

  1. Since our network is trained on Monte Carlo-defined jet labels, consistency requires evaluating interpretability on the same labels; otherwise, the comparison would not be meaningful. We have clarified this in Sec. 2. Our approach is agnostic to the jet definition and could equally be applied in semi-supervised or data-driven labeling scenarios. We also stress that we verified robustness by comparing Pythia and Herwig and found consistent latent directions despite differences in absolute performance. We now state clearly that our results are a deconstruction of what the network learns in a given simulator, not a claim about a unique, theory-defined notion of quark/gluon in nature.

  2. We have clarified our selection principle: Eq. 8 contains a minimal set of observables known to perform well in QG tagging and have been studied extensively in the past. In particular, in the context of ML-enhanced tagging techniques. Hence, we first reanalyzed the same set of observables and found that they align well with the three leading PCA directions (multiplicity/diversity, radial structure, fragmentation) encoded in the network's latent space. We later indeed identify other observables that are equally important, if not more so. We also tested energy flow polynomials, but in our setup, they performed worse than the chosen and presented variables, which we now explicitly state in the manuscript. EFPs can be mapped onto the latent space, but two issues arise: i) the latent space is not guaranteed to be IRC safe, which means EFPs are not able to reconstruct all latent directions, and ii) we would need many composite EFPs to represent a principal component. However, this is no longer human-interpretable. We now added a section explaining these issues.

  3. We agree that Pearson correlations capture only linear dependence. This choice was deliberate since PCA itself resolves only linear structure. We now make this explicit and complement it with a mutual information study that confirms the same qualitative axes. Through this additional study, we raised concerns about the interpretability of mutual information and explained that the results align with our previous interpretation, further justifying our initial statement based on Pearson's correlations.

  4. They have a physical meaning in that they illustrate which combination of known physical observables the network uses to construct a decision boundary, and that these linear combinations are aligned with the latent axes. Eq. 10 encodes “multiplicity + diversity,” while Eq. 13 isolates jet width at fixed multiplicity, which is explicitly stated in the text. The linear combination increases the Pearson correlation, showing that, for example, PC1 is really more than just multiplicity.

  5. We appreciate the concern and clarified that our goal is not to reproduce perturbative QCD distributions, but to identify which observables the network encodes in its decision boundary. This complements theoretical predictions: QCD determines distributions, while our methods reveal feature usage.

  6. We corrected the caption to clearly state what is shown.

  7. It is explicitly stated that Eq. 18 applies the orthogonalization from Eq. 17 to isolate the PC3-aligned component, making the purpose of this formula clear: finding which linear combination of known physical observables is aligned with PC3 and thus is encoded in the latent space of the network.

  8. We agree that the covariance loss only enforces linear decorrelation, but this is precisely what we want. Completely decorrelating jet observables is neither sensible nor interpretable. Each latent direction can still contain complex combinations of jet observables, albeit nonlinearly. The covariance penalty is only to help us categorize classes of observables. What we want to avoid is seeing the same Pearson correlation between one observable and all three latent directions, while keeping in mind that our high-level observables are already nonlinearly correlated. We clarified this in Sec. 3.2 now.

  9. We clarified in the caption that the red-blue gradient encodes the respective feature value. “high” and “low” are the respective maxima and minima of each observable. We refrained from explicitly showing histograms of observables, as they have already been shown in many papers before.

  10. see after point 11.

  11. The paragraph in question (p. 12) makes a specific methodological point: the counter-intuitive SHAP attribution arises because SHAP assumes feature independence and marginalizes over correlated inputs. In practice, wpf​ is correlated with npf​: low wpf​ appears in both quark jets (low multiplicity) and some gluon jets (high multiplicity). The classifier correctly learns this joint structure, but SHAP ignores it, producing a misleading attribution. Documenting such pitfalls is precisely the kind of methodological caveat we intend this paper to highlight. More generally, we emphasize that this limitation is not inherent to Shapley values themselves, but to SHAP and its independence assumption. Our decorrelated-input analysis shows that, once correlations are removed, the attributions align with physical intuition. We included this “by-hand” decorrelation only to illustrate the source of the problem; we agree it does not scale to high dimensions and do not recommend SHAP for such cases. We believe that showing both the failure mode and a partial remedy remains valuable for practitioners who might otherwise misinterpret SHAP results in collider applications.

  12. For us, interpretability means we can identify the dominant physical observables encoded in the network's latent space. We have now clarified this definition in our paper. Again, the interpretation of the identified physical observables themselves is QCD and independent of this analysis. Our task is not to redefine how these physical observables should be interpreted using ML.

  13. We clarified that ΔC quantifies calibration, which complements the AUC. Unlike AUC, it is not invariant under monotonic transformations; hence, it can change even when discrimination power does not. However, to have a statistically well-defined classifier, it needs to be calibrated. We also checked robustness with respect to binning, finding negligible variation. We reiterate this clearly in Sec. 5.1.

  14. The appearance of tanh reflects that the decision function needs to be bounded within [0,1]. Interpretability here means identifying the observables the network uses, not reproducing distributions, which should be done with QCD, as the referee correctly states. The formulas give us an analytic description of the network, not QCD results.

  15. We clarified that by “new, refined combinations” we mean combinations such as r_lambda or SPID+npf, which are known observables but not typically used directly in quark/gluon tagging. The novelty lies in showing the network actually encodes them in effective combinations for discrimination.

  16. The appendices provide the full PCA and symbolic-regression scans underlying the main-text figures. We now mention this in the corresponding sections, so the reader understands they are included for completeness and reproducibility.

---

## Round 1 · Referee Report · Anonymous (Referee 2) · 2025-10-17

Report

The manuscript titled "How to Deep-Learn the Theory behind Quark-Gluon Tagging" presents a thoughtful and timely XAI study of quark–gluon tagging, combining PCA/DLC analyses, SHAP attributions, and symbolic regression to connect ParticleNet-Lite to physics-motivated observables. The results are promising—e.g. clear latent directions (multiplicity, radial profile, fragmentation) and a compact multi-observable surrogate. Before publication, however, I would like to see a few points regarding clarity/notation, reproducibility and robustness. If the authors address the specific points listed below, I would be happy to recommend publication. My overall recommendation at this stage is major revision.

Requested changes

  1. Please report the exact ParticleNet-Lite baseline metrics directly in Fig. 3 or in the accompanying text: the AUC and the background rejection at 30% efficiency. Indicating both values will make comparisons to the PCA/DLC/Symbolic regression variants immediately clear.

  2. Please fix Eq. (7) for consistency with the zero-centering step. It should read Z=(X-mu_X) V (since Eq. (5) uses mean-centered features), unless you explicitly redefine X to already denote mean-centered data. In the latter case, please state this redefinition clearly before Eqs. (5)–(7).

  3. Please state explicitly how the class labels map to the network output. For clarity to non-experts, add a sentence early in Sec. 2 (or where the classifier is introduced) that the model outputs the quark probability (quark = 1, gluon = 0), i.e., scores closer to 1 indicate quark-like jets and scores closer to 0 indicate gluon-like jets.

  4. Please improve the visual distinguishability in Figs. 3 and 10. Several curves are rendered with similar shades of blue, which makes them hard to tell apart.

  5. Page 6 Typo "correllation"

  6. Page 8 Please fix the notation in Eq. (18): replace S with S_{frag} to avoid confusion with other entropies.

  7. Please add the label “reconstruction” next to x' in Fig. 6 to make the reconstruction head explicit.

  8. Please remove the extra (') in Table 1.

  9. Please add a brief stability analysis of the DLC latent vectors across random seeds. In particular, report whether the identities (and ordering) of z1, z2, z3​ are consistent under retraining, and how stable their correlations—and the resulting ranking—with respect to key observables are. This will clarify whether the physical interpretation assigned to z1, z2, z3​ is robust.

  10. Please add the necessary implementation details to ensure reproducibility of both the DLC and the symbolic-regression pipelines. Alternatively, a public repository containing training scripts and configs would fully address this point.

  11. If I understand correctly, Fig. 8 is a scatter plot? If so, please state explicitly in the Fig. 8 caption that it is a scatter plot of SHAP values and the features are ordered by the mean absolute SHAP value across events. If I have misunderstood the plot or the ranking criterion, please clarify what a dot represents and the exact ordering rule.

  12. As a suggestion, please consider adding, in the Symbolic Regression section, score‐distribution plots on the test set for each optimal formula (1D/2D/7D). Overlay the quark and gluon histograms and compare them on the same axes with the corresponding MLP and ParticleNet-Lite outputs. These plots would reveal where the analytic formulas diverge from the learned models (e.g., in tails).

  13. As a suggestion aligned with your Outlook, maybe you could include a lightweight, reproducible benchmark where the optimal symbolic-regression formulas (1D/2D/7D) are evaluated as fast surrogates on an independent quark/gluon dataset and compared to ParticleNet-Lite in inference latency, memory usage and AUC.

  14. The title feels overstated. Unless Eq. (31) can be endowed with a clear, physics-grounded interpretation, I recommend softening the claim in the title.

  15. As EFP (Energy Flow Polynomials) provide a mathematically complete and IRC-safe basis for jet observables and have been used in prior work for interpretable or surrogate models, it would be valuable if the authors could comment on the choice of not using an EFP basis for symbolic regression.

Recommendation

Ask for minor revision

  • validity: good
  • significance: good
  • originality: good
  • clarity: high
  • formatting: excellent
  • grammar: excellent

Author:  Ramon Winterhalder  on 2025-12-23  [id 6183]

(in reply to Report 2 on 2025-10-17)
Category:
answer to question
reply to objection

We would like to sincerely thank you for the careful reading of our manuscript and for providing valuable comments and suggestions. We have carefully revised the manuscript according to the report.. Below we provide a point-by-point response to each comment:

  1. We now report exact baselines in Fig 3.
  2. This equation is now fixed.
  3. This section now contains a sentence clarifying this.
  4. We have optimized the choice of color-palette for these figures.
  5. This typo is fixed.
  6. This equation is fixed now.
  7. The label is added to Fig.6.
  8. This is fixed.
  9. We have added a stability analysis for the DLC.
  10. We have added tables containing hyperparameters for both the SR and the DLC.
  11. We have changed the caption to make the plot clearer.
  12. We have added a score distribution plot of the 7D optimal formula across different runs to show its robustness, and we have compared it to the MLP and ParticleNet-Lite baselines.
  13. While we appreciate this suggestion, we want to leave this test for future work. The surrogate equations were a proof-of-concept to reveal what a classifier is doing, rather than primarily a more robust framework.
  14. We have changed the title to be more precise about the actual scope of our paper.
  15. We added a section explaining the shortcomings of EFPs in this type of analysis and also included them in the PCA analysis, which directly explains our decision not to use them for symbolic regression.

---

## Round 1 · Referee Report · Ezequiel Alvarez (Referee 3) · 2025-10-27

Strengths

1- Addresses the question of simulations validity in QCD 2- Obtains analytic formulas for an instantiation of a simulation

Weaknesses

1- Paper objectives are not clear

Report

The authors study the interpretability of a ParticleNet‑Lite quark–gluon tagger trained on low‑level jet constituents. They: - Analyze the multidimensional pooled latent representation using PCA and a disentangled latent classifier (autoencoder + classifier); - Assess feature importance via SHAP and discuss pitfalls from correlated inputs; - Approximate the classifier with compact analytic formulas via symbolic regression (PySR). They identify three leading latent directions aligned with familiar physics: (i) multiplicity/particle‑type diversity, (ii) radial energy flow, and (iii) fragmentation/energy dispersion. They propose decorrelated observables to mitigate correlation issues for SHAP and present analytic formulas that mimic a tagger trained on selected observables. Robustness is partially explored by testing PCA directions across Pythia and Herwig.

After reading the paper and going through some of its sections many times, I was still with the feeling that something is missing. In my understanding the reason for this is that it is not fully clear the objective of the paper.

From my perspective, I see two important points

1) Lower the dimensions to extract the right info: This is a common and useful –if not crucial– pattern in ML. Given that non‑perturbative QCD cannot be modeled perfectly, it is reasonable to seek low‑dimensional summaries—within a given simulator instantiation—that capture what the tagger needs. 2) Proposing analytic formulas that surrogate the network within one simulation setup.

The purpose of (2) is not sufficiently clear in the manuscript. This is not necessarily a problem if the authors argue convincingly for future utility. For example, such analytic formulas could serve as physics‑informed priors or mean functions in a Bayesian framework (e.g., estimated via Hamiltonian MC). One could promote the formula coefficients to parameters with weakly informative priors and let the Bayesian inference update them on data, or place a Gaussian‑process perturbation around the formula and infer hyperparameters from data.

However, to pursue these or related goals, a more robust assessment of simulation independence is needed, especially for the analytic formulas. I encourage the authors to focus on the robustness of the formulas themselves.

Concrete suggestions for strengthening robustness: - Go beyond a binary Pythia vs Herwig comparison by including multiple Pythia tunes. It is valuable not only to show case where tune dependence is mild, but also to map out the limits of robustness. - Cross‑tune transfer: train the symbolic formulas (and the small ML surrogates they approximate) on tune A and evaluate them on tune B; then vary the tunes until AUC/rejection/calibration degrades noticeably. This will quantify the range over which the formulas are reliable and where they break down.

Future‑work idea (optional, but illustrative of utility). If formulas trained on tune A degrade on tune B, compare “use‑as‑is on B” vs “Bayesian recalibration on B” in which the coefficients are inferred on B after seeing the data, while simultaneously using the formula for tagging. One can expect the latter to perform substantially better; this would showcase the practical value of having a compact analytic surrogate.

Recommendation I recommend publication after major revision.

Requested changes

1- Clarify and sharpen the paper’s objectives early in the introduction. 2- Tone down the title and several statements that currently oversell the results; align claims with demonstrated evidence. 3- Substantially expand the robustness study of the analytic formulas across simulations: include multiple Pythia tunes and explicitly identify the regime where performance meaningfully degrades (AUC, or also rejection or calibration), so readers can gauge the limits of simulation independence.

In my understanding, these changes would make the paper’s aims explicit, better calibrate the claims, and provide the robustness evidence needed for the proposed use of analytic surrogates.

Recommendation

Ask for major revision

  • validity: good
  • significance: good
  • originality: high
  • clarity: good
  • formatting: excellent
  • grammar: perfect

Author:  Ramon Winterhalder  on 2025-12-23  [id 6182]

(in reply to Report 3 by Ezequiel Alvarez on 2025-10-27)
Category:
answer to question
reply to objection

We would like to sincerely thank you for the careful reading of our manuscript and for providing valuable comments and suggestions. We have carefully revised the manuscript according to the report. Below we provide a point-by-point response to each comment:

  1. The analytic expressions are solely there to understand the decision boundary of a trained network. Although there are many interesting applications, it was not our intention to develop a complementary framework to Neural Networks, but rather to use these expressions to understand the networks. That said, questions of robustness should focus on the network we approximate in the first place, since symbolic regression only replicates the network's performance. We have now revised the introduction and the symbolic regression section to make our intentions clearer. Additionally, to demonstrate some robustness, we also evaluated the analytic formula on the Herwig dataset.

  2. The aim of this study was not to derive a robust expression, but to understand what a network learns and to express this analytically, especially on a standard benchmark dataset. To still address the question of robustness, we now include a comparison of the formula's performance on the Herwig dataset. The robustness questions should be addressed for the MLP baseline, which is beyond the scope of this paper. We now explicitly state that the SR formula will only be as robust as the network it was trained on.

  3. Thank you very much for this interesting ansatz. We will consider this as future work and not include it in this paper.

---

## Editorial Decision

resubmitted